

**The CU 2D-MAX-DOAS instrument - part 2: Raman Scattering Probability Measurements and**
**Retrieval of Aerosol Optical Properties**
Ivan Ortega[1,2], Sean Coburn[1,2], Larry K. Berg[3], Kathy Lantz[2,4], Joseph Michalsky[2,4], Rich Ferrare[5],
Johnathan Hair[5], Chris Hostetler[5], and Rainer Volkamer[1,2]
[1]Department of Chemistry and Biochemistry, University of Colorado, Boulder, CO, USA
[2]Cooperative Institute for Research in Environmental Sciences (CIRES), Boulder, CO, USA
[3]Pacific Northwest National Laboratory, Richland, WA, USA
[4]Global Monitoring Division, Earth System Research Laboratory, NOAA, Boulder, CO, USA
[5]NASA Langley Research Center, Hampton, VA, USA
*Correspondence to:* Rainer Volkamer (rainer.volkamer@colorado.edu)
**Abstract**
The multiannual global mean of aerosol optical depth at 550 nm ($AOD_{550}$) over land is ~0.19, and that
over oceans is ~0.13. About 45% of the Earth surface shows $AOD_{550}$ smaller than 0.1. There is a need
for measurement techniques that are optimized to measure aerosol optical properties under low AOD
conditions. We present an inherently calibrated retrieval (i.e., no need for radiance calibration) to
simultaneously measure AOD and the aerosol phase function parameter, *g*, based on measurements of
azimuth distributions of the Raman Scattering Probability (RSP), the near-absolute Rotational Raman
Scattering (RRS) intensity. We employ Radiative Transfer Model simulations to show that solar
azimuth RSP measurements are insensitive to the vertical distribution of aerosols, and maximally
sensitive to changes in AOD and *g* under near molecular scattering conditions. The University of
Colorado two dimensional Multi-AXis Differential Optical Absorption Spectroscopy (CU 2D-MAX-
DOAS) instrument was deployed as part of the Two Column Aerosol Project (TCAP) at Cape Cod,
MA, during the summer of 2012 to measure direct sun spectra, and RSP from scattered light spectra at
solar relative azimuth angles (SRAA) between 5˚ and 170˚. During two case study days with (1) high
aerosol load (17 July, $0.3 < AOD_{430} < 0.6$) and (2) near-molecular scattering conditions (22 July,
$AOD_{430} < 0.13$) we compare RSP based retrievals of $AOD_{430}$ and *g* with data from a co-located CIMEL
sun photometer, Multi-Filter Rotating Shadowband Radiometer (MFRSR), and airborne High Spectral
Resolution Lidar (HSRL-2). The average difference (relative to DOAS) for $AOD_{430}$ is: $+0.012 \pm 0.023$





(CIMEL), -0.012 ± 0.024 (MFRSR), -0.011 ± 0.014 (HSRL-2), and +0.023 ± 0.013 (CIMEL$_{AOD}$ −
MFSRS$_{AOD}$); and yields the following expressions for correlations between different instruments:
DOAS$_{AOD}$ = -(0.019 ± 0.006) + (1.03 ± 0.02)·CIMEL$_{AOD}$ ($R^2$ = 0.98), DOAS$_{AOD}$ = -(0.006 ± 0.005) +
(1.08 ± 0.02)·MFRSR$_{AOD}$ ($R^2$ = 0.98), and CIMEL$_{AOD}$ = (0.013 ± 0.004) + (1.05 ± 0.01)·MFRSR$_{AOD}$
($R^2$ = 0.99). The average $g$ measured by DOAS on both days was 0.66 ± 0.03, with a difference of
0.014 ± 0.05 compared to CIMEL. Active steps to minimize RSP in the reference spectrum help to
reduce the uncertainty in RSP retrievals of AOD and $g$. As AOD decreases, and solar zenith angle
(SZA) increases the RSP signal-to-noise ratio increases. At AOD$_{430}$ ~ 0.4 and 0.10 the absolute AOD
errors are ~0.014 and 0.003 at 70$^°$ SZA, and 0.02 and 0.004 at 35˚ SZA. Inherently calibrated, precise
AOD and $g$ measurements are useful to better characterize the aerosol direct effect in urban polluted
and remote pristine environments.

**1.  Introduction**
Atmospheric aerosol particles play a key role in the energy balance of Earth's atmosphere (IPCC,
2013). The aerosol optical depth (AOD), defined as a vertical integral of the aerosol extinction
coefficient from the earth surface to the top of the atmosphere, is an important input to assessments of
how the atmospheric aerosol burden affects the budget of incoming solar radiation in global climate
models (Hansen et al., 2002; Chung et al., 2005; McComiskey et al., 2008). McComiskey et al. (2008)
studied the sensitivity of aerosol direct radiative forcing using representative uncertainties in currently
established methods to measure aerosol optical properties. For a typical AOD uncertainty of 0.01 (best
case scenario expected for newly calibrated ground based radiometric instrument in the visible spectra
range; Eck et al., 1999; Holben et al., 1998), the error in the aerosol direct forcing is about 0.6 W·m$^{-2}$
(top of the atmosphere) and 1.3 W·m$^{-2}$ (surface) for a solar zenith angle (SZA) of 45˚ (McComiskey et
al., 2008). The multiannual global mean AOD$_{550}$ estimated from satellites find that about 28% and 43%
of the land surface, and 15% and 46% of the ocean surface have AOD ≤ 0.05, and ≤ 0.1 (Remer et al.,
2008); current ground based networks capture frequent AOD values below 0.15 (Holben et al., 2001;
Augustine et al., 2008; Michalsky et al., 2010; Mao et al., 2014). Low AOD conditions are projected to
be more prevalent in the future (Westervelt et al., 2015). Under these conditions measurements of AOD
with higher accuracy and precision are even more desirable.
Traditional AOD measurements often employ radiometric calibrated instruments, e.g., CIMEL sun
photometer (Holben et al., 1998) and multifilter rotating shadowband radiometer (MFRSR) (Harrison et



64 al., 1994). In general, the retrieval of AOD is estimated based on the extinction of the direct sun

65 irradiance measurements. The quality of such measurements is improved under high AOD and cloud

66 free conditions. On the other hand, under molecular scattering conditions, i.e., $AOD_{430} < 0.13$ (=

67 Rayleigh scattering extinction under overhead sun conditions), the measurements become subject to

68 higher relative uncertainties (Holben et al., 1998). Holben et al. (1998) pointed out that the error in

69 AOD by means of solar sky brightness (scattering) in the solar aureole region may be lower than

70 traditional direct sun extinction methods. However, to our knowledge, this has not previously been

71 exploited in measurements to date.

72 Multi-AXis Differential Optical Absorption Spectroscopy (MAX-DOAS) can simultaneously retrieve

73 trace gases and aerosol optical properties (Hönninger et al., 2004; Wagner et al., 2004; Frieß et al.,

74 2006; Clémer et al., 2010). The MAX-DOAS technique relies on spectrally resolved solar scattered

75 light measurements at several elevation angles (EA), defined between the horizon and zenith

76 (Hönninger et al., 2004). The retrieval approach does not require radiometric calibration, and the trace

77 gases and aerosol optical properties are measured relative to a reference spectrum, typically recorded in

78 the zenith. Measurements at low EA have maximum sensitivity in the lowermost part of the

79 atmosphere. More recently, two dimensional (2D) MAX-DOAS has been shown to be a promising

80 technique to measure the trace gas variability around the measurement site from scattered light spectra

81 at different Azimuth Angles (AA), defined relative to North (Wang et al., 2014; Ortega et al., 2015a).

82 The University of Colorado (CU-) 2D-MAX-DOAS instrument has demonstrated range resolved

83 measurements of $NO_2$ and oxygenated hydrocarbons from azimuth scans at low EA. The spatial scale

84 probed by 2D-MAX-DOAS closely resembles the grid-cell size of atmospheric models and satellite

85 pixels, and can be used to systematically characterize chemical gradients under inhomogeneous

86 conditions (Ortega et al., 2015a).

87 In this paper we exploit solar azimuth scattered light and direct-sun measurements to assess aerosol

88 column properties using solar almucantar measurements. The information content regarding aerosol

89 properties using this geometry has been discussed in detail for radiance measurements with single

90 wavelength channel detectors elsewhere (Box and Deepak, 1979; Nakajima et al., 1983; Kaufman et al.,

91 1994; Bohren and Huffman, 1998; Dubovik et al., 2000). We use solar almucantar scans in combination

92 with hyperspectral measurements, and describe a new retrieval scheme to estimate $AOD_{430}$ and aerosol

93 phase functions (simplified by $g$, Henyey-Greenstein approximation) based on quantitative analysis of

94 the Rotational Raman Scattering (RRS) by atmospheric molecules (Ring effect) (Grainger and Ring,





1962; Chance and Spurr, 1997). RRS causes "filling-in" of the solar Fraunhofer lines, and has to be
taken into account to accurately estimate absorption of trace gases using passive DOAS techniques
(Platt and Stutz, 2008). The quantitative analysis of RRS by DOAS was introduced by Wagner et al.
(2004, 2009a) with the so-called "Raman Scattering Probability" (RSP) (the probability that a detected
photon has undergone a rotational Raman scattering event). Under cloud free conditions the AOD has a
strong effect on the RSP, which further exhibits a high dependency on the solar relative azimuth angle
(Wagner et al., 2009b; Wagner et al., 2014). To the best of our knowledge, there has been no previous
measurement of AOD and *g* using almucantar scans of RSP by MAX-DOAS.

## 103  2. Experimental

### 104  2.1     The TCAP field campaign

The first phase of the Department of Energy (DOE) Two-Column Aerosol Project (TCAP) field
campaign took place at Cape Cod, MA during the summer of 2012 (Berg et al., 2015). TCAP was
designed to provide a comprehensive characterization of the aerosol direct and indirect effects under
urban emission influences near the east coast of North America (over Cape Cod, MA), and to contrast it
with observations in   pristine conditions over the Atlantic Ocean. An extensive set of aerosol
measurements were conducted aboard two research aircrafts (DOE G-1 and NASA B200) and with the
DOE Atmospheric Radiation Measurement (ARM) ground Mobile Facility (located over Cape Cod,
MA, U.S.); for details see Berg et al. (2015). The CU 2D-MAX-DOAS was deployed at the ARM
ground site from 1 July to 13 August 2012 to test its innovative capabilities to measure aerosol optical
properties and trace gases simultaneously with a single instrument. Here, we focus primarily on 22 July
2012 due to its low AOD and cloud free conditions, and the available complementary data (Berg et al.,
2015; Ortega et al., 2015b). The retrieval approach is also applied for a high AOD case study on 17
July 2012. The TCAP data set provides an excellent opportunity to evaluate the robustness of the RSP-
based retrieval approach and to compare the products with independent instruments. Table 1 and
Section 2.6 present other measurements and products used in this work.

### 121  2.2     2D-MAX-DOAS measurements

The 2D-MAX-DOAS telescope and detection system has been described in detail elsewhere (Ortega et
al., 2015a). The angles defining the geometry of measurements are illustrated in Fig. 1. The CU 2D -
MAX-DOAS instrument conducts measurements in three different modes: (1) off-axis scan where
several elevation angles (EA) and zenith are used with a fixed azimuth angle (AA) relative to north; (2)





almucantar scan, where solar scattered photons are collected using any EA for multiple solar relative
azimuth angles (SRAA). To further enhance the aerosol information content and estimate uniformity
(homogeneity) around the measurement site the almucantar scan is carried out on the left and right side
of the solar disc. Up to now, this particular geometry has not been used with MAX-DOAS, however it
is widely used by the CIMEL sun photometer using single wavelength channel detectors at solar
elevation (EA = 90˚ - SZA) (Holben et al., 1998); and (3) direct sun observations, which inherently
minimize RSP.
The 2D-MAX-DOAS instrument deployed during TCAP at the ARM Mobile Facility consisted of three
synchronized spectrograph/detector units located indoors in a temperature controlled sea container, the
control measurement laptop, and the 2D telescope located outdoors. The telescope was deployed on top
of one seatainer (~45 m ASL) providing an unobstructed view close to the horizon in a ~360˚ azimuth
view. The only small obstruction in the azimuth scan was an independent sampling inlet pillar located
in the middle of the seatainer. The light collected with the telescope was focused onto a single
CeramOptics  25 m x 1.0 mm silica mono fiber coupled to a tri-furcated fiber bundle connected to three
Ocean Optics (QE65000) spectrometers collecting solar light between 300 and 631 nm with a spectral
resolution between 0.4 - 0.6 nm (FWHM). The same spectrometer system was used in the remote
Pacific Tropical Ocean for the detection of glyoxal (Sinreich et al., 2010). The electronic rack
containing the spectrograph/detectors was temperature controlled (34 ˚C, 0.005 ˚C peak to peak
variation) and CCDs cooled to -30 ˚C to minimize dark current.
**2.2.1   Configuration of the azimuth scan**
The instrument was configured to conduct measurements of direct sunlight, and scattered sunlight using
a sequence of EA and AA pairs described in Table 2. The off-axis scan consisted of 7 EAs and zenith,
and spectra were recorded using an integration time of 1 min at each angle alternating South and North
AAs (total acquisition time of 16 – 17 min). This specific geometry was used in order to know the
effect of elevated aerosol layers in the apparent absorption of the oxygen collision complex ($O_2$-$O_2$) as
seen by the 2D-MAX-DOAS (Ortega et al., 2015b). At the end of the EA scan, the almucantar scan was
implemented with an integration time of 1 s with 70 angles relative to the sun in steps of 5˚ up to 180˚
on the left and right sides of the solar disk at solar EA. The almucantar scan was repeated for a fixed
EA of 45˚. The total acquisition time of the azimuth scan was 2 - 3 min. In this work, we focus only on
the almucantar scan at solar EA. The full measurement cycle between EA and almucantar scans took
about 20 min and was repeated sequentially. The initial solar almucantar alignment procedure to
achieve pointing accuracy better than the motors internal encoder resolution (0.17˚) is described in
detail by Ortega et al. (2015a). Briefly, the initial alignment is carried out in the field by measuring
rapid (1 s, integration time) solar scattered spectra with several small SRAA (usually -5˚ < SRAA < 5˚,
negative SRAAs are left and positive values are right side of the sun). The alignment is achieved when
measurements of intensities (in counts·s$^{-1}$) on the right and left sides present symmetry and the offset
estimated with a Gaussian fit of the intensities at the center of the sun's disk is small (< 0.17˚) and
accounted in the software. To avoid saturation of the detector, this alignment procedure was performed
below and above the sun position (see Fig. 2 in Ortega et al., 2015a). The telescope field of view (FOV)
of this viewing port was determined by introducing light into the fiber retrospectively from the exit
side, and the divergence of the light after exiting the telescope was evaluated to have a full opening
angle of 0.6˚ in agreement with the theoretical FOV based on the experimental field setup.

### 168   2.2.2   Direct sun mode

During the first phase of TCAP, for cloud-free days, direct sun spectra were recorded periodically with
a total integration time of 2 - 4 min. In order to reduce the intensity of the direct sun beam and avoid
saturation of the detector the light is collected via an integrating sphere with a diameter of 2.54 cm. The
sphere also serves for correcting pointing inaccuracies and atmospheric lens effects (Herman et al.,
2009). To minimize the contribution of solar scattered photons in the direct sun mode a black anodized
collimator tube with a full opening angle of 2.9˚ was used. A sketch of the optics housed integrating the
direct sun and azimuth ports is shown in Fig.1 in Ortega et al. (2015a). The custom software developed
in LabView uses the exact coordinate location and heading (defined as zero corresponding to true
north) to operate the 2D telescope. This information is used as Euler angles to correct the astronomical
solar position and locate the sun in the sky. This step is similar to the crude alignment of advanced solar
trackers, which apply active imaging of the solar disk for precise pointing (Gisi et al., 2011; Baidar et
al; 2015). We do not aim to track the sun in this work. The purpose of the direct sun mode is to obtain
spectra that are near-free of RRS, and use these direct sun spectra as reference spectra in the retrieval of
RSP. To assess pointing accuracy of the direct sun observation we use the solar azimuth scan alignment
as explained in section 2.2.1.

### 185   2.3   DOAS retrieval of differential RSP and intensities

The main products retrieved with the solar azimuth scan are the non-calibrated spectral intensities



($I_{norm}$) and the strength of RRS by atmospheric molecules (RSP). The spectra intensities were corrected
by electronic offset and dark current, and the number of CCD-pixel counts were normalized by the
integration time (units of counts·$s^{-1}$) at a certain wavelength ($\lambda$). These normalized $I_{norm}$ are used for
quality assurance of homogeneity and to calculate pointing accuracy only. The differential RSP (dRSP;
differential with regards to the amount contained in the reference spectrum) was measured by its
specific narrow band signatures (< 1 nm) at UV-Vis wavelengths (Fig. 2), which are separated well
from broadband molecule and aerosol extinction using the DOAS method (Platt and Stutz, 2008). We
follow the retrieval strategy introduced in Wagner et al. (2009a) and apply the DOAS settings from
Wagner et al. (2009b) to retrieve the RSP in the fitting window of 426 – 440 nm. The only atmospheric
cross section absorber adjusted to the spectrometer resolution that is included in the analysis is $NO_2$
(Bogumil et al., 2003). A third order polynomial is fitted to account for broad band spectral structures.
A direct sun spectrum recorded at low SZA (28˚) on 22 July 2012 is used as reference spectrum to
evaluate the dRSP in the azimuth scan mode. The Ring cross section is calculated from the respective
sun-observation spectrum using the DOASIS software (Kraus, 2006), which then is normalized by
removing the continuum component with a third order polynomial high pass filter (Wagner et al.,
2009a). The spectra were analyzed using the WinDOAS software package (Fayt and Van Roozendael,

203    2001).

An example of the DOAS fit analysis is shown in Fig 2. Systematic errors in the retrieval of dRSP were
quantified by means of sensitivity studies. The sensitivity of the DOAS settings were explored by
changing the wavelength intervals and polynomial orders in a similar way as performed by Vogel et al.
(2013). These sensitivity tests reveal a remarkable stability towards changing the DOAS fitting window
using different polynomial orders (see Fig. S1; difference < 5%; the same analysis in the UV, Fig. S2,
yields two times greater DOAS fit error and root mean square (RMS) due to the smaller signal to noise
ratio achieved with 1 s integration time). The typical value of the dRSP fit error is ~0.0018, calculated
internally in WinDOAS as the standard deviations on the retrieved dRSP; it interestingly does not
depend strongly on the SRAA. We adopted this uncertainty in the final error propagation of the aerosol
optical properties (see section 3.3.1).
Figure 3 shows typical examples of the measurements of dRSP and $I_{norm}$ obtained with the solar
azimuth scan (mode 2) for three different SZAs. The SRAA scan is from -180˚ (left side of the sun) to
+180˚ (right side of the sun). The dRSP decreases for small SRAAs due to fewer scattering events by
molecules and a dominant aerosol forward scattering. On the other hand the $I_{norm}$ increases for small





SRAAs due to the strong probability of aerosol scattering in the forward direction. The second
important aspect is the SZA dependency. Previous studies have established the relationship between the
SRAA, SZA and the effective aerosol scattering angles (Nakajima et al., 1996; Torres et al., 2013). In
general, the information content of the azimuth scan is maximized by using high SZAs. The maximum
dRSP values (corresponding to a minimum $I_{norm}$) are shown at SRAA of 100˚ (for SZA = 66˚), which
indicates to some degree the high sensitivity to aerosol scattering processes (aerosol phase function).
The dRSP decreases for low SZA (blue circles).
**2.4    Radiative Transfer Simulations**
We use the full spherical Monte-Carlo atmospheric radiative transfer model (McArtim) (Deutschmann
et al., 2011) to simulate and interpret the measurements. McArtim has been successfully tested and
compared with other radiative transfer models (Wagner et al., 2007). McArtim simulates atmospheric
photon transfer using the optical properties described by several input parameters such as vertical
profiles of pressure, temperature, and aerosol extinction characterized with aerosol phase functions,
typically represented by $g$, and single scattering albedo (SSA). McArtim calculates the absolute RSP
using the fraction of scattering events that have presented RRS (inelastic scattering). Reflection at the
surface is characterized with the surface albedo (SA) and is treated as Lambertian. The modeled RSP
from McArtim has been previously characterized and used in several studies (Wagner et al., 2009b;
Wagner et al., 2010; Wagner et al., 2014). Several general input parameters are required and kept
constant in the forward modeling. An altitude grid of 100 m up to 10 km, 200 m up to 50 km, and 5 km
up to 100 km was used. The FOV was set to 0.6˚ (similar to the full opening angle of the telescope, see
section 2.2.1). The wavelength chosen to forward model the RSP is 430 nm representing the middle
wavelength of the fitting window and characteristic Ca-lines of the Fraunhofer spectrum (Wagner et al.,
2010). In this section we describe the different sensitivity studies that were performed in order to
understand the effect of aerosol optical properties in the measured RSP using the solar azimuth scan
geometry. For the sensitivity studies we use the pressure, temperature, and RH profiles taken from the
U.S Standard Atmosphere. We have adopted the geometry of typical 2D-MAX-DOAS measurement
taken from TCAP, i.e., similar SRAA angles (-180 to +180˚) and SZA ranges.

**2.4.1   Sensitivity of RSP to aerosol distribution**
Figure 4A presents the effect of AOD on the simulation of RSP in the azimuth scan for a single SZA
(70˚). Additional input properties are SSA = 0.98, $g$ = 0.70, SA = 0.05, and homogeneous extinction



height of 1.5 km. As expected (see Fig. 3), the RSP decreases and the radiance increases for angles
close to the sun (see Fig. 4A). In general, the RSP decreases with increasing AOD due to the decrease
of molecular scattering and higher probability of aerosol elastic scattering (see section 3.3 for further
analysis regarding maximal low AOD information). Figure 4A also shows that the maximum RSP is at
about $90 - 100°$, in agreement with our measurements at similar SZA (see Fig. 3A). Figure 4B shows
the sensitivity of the RSP with respect to the aerosol extinction vertical distribution while keeping the
AOD constant at 0.2 (additional parameters are the same as before). Several homogeneous extinction
vertical profiles from altitudes of 0.5 to 2.0 km are studied as well as a case of aerosol extinction aloft,
assuming maximum extinction at 2.8 km with a width of 2.8 km. Similar results are obtained also at
small SZA (see Fig. S3). It is clear that the aerosol extinction vertical distribution does not play a
significant role in the simulation of RSP in the azimuth scan. Systematic elevated aerosol extinction
layers were identified during TCAP (Berg et al., 2015; Ortega et al., 2015b). Previous studies have
shown that RSP is primarily sensitive to the AOD (Wagner et al., 2009b), and recognize the value of
measurements at small SRAA to obtain information about $g$ (Holben et al., 1998; Wagner et al., 2009b).
The sensitivity studies in Figs 3, 4, and in the supplement confirm that the RSP does not depend on the
aerosol vertical distribution.  Hence, the RSP is suitable to characterize column properties of AOD and
the aerosol phase function, $g$. The elevated aerosol layers documented by Berg et al. (2015) during
TCAP hence are captured, and do not present a limitation for this work.
**2.4.2   Sensitivity to $g$, SSA, and SA**
The second sensitivity study aimed to understand the effect of $g$, SSA, and SA. Figure 5 shows the
results of the RSP and Sun-normalized radiances, $R_{norm}$ ($sr^{-1}$), defined internally in McArtim as the ratio
of the radiance ($W·m^{-2}·sr^{-1}$) of the geometry indicated to the solar irradiance ($W·m^{2}$) using a
homogeneous aerosol extinction profile with an AOD of 0.2 (box height of 1.5 km), and SZA of 70°.
The asymmetry parameter, $g$, has the strongest effect on the RSP, especially for SRAA < 40°. The
importance to the RSP of the geometry of measurements and its qualitative sensitivity towards aerosol
phase functions was identified by Wagner et al. (2009b) using three different fixed azimuth directions.
The RSP does not show a significant variability among different SSA, i.e., aerosol composition,
however the sun-normalized radiances show some sensitivity among all SRAA, especially with angles
close to the sun where variability of up to 10% are found. A similar sensitivity study was shown in
Frieß et al. (2006). The SA does not play a significant role in the simulation of either the RSP or
radiances. Further discussion of the phase functions is presented in section 3.3.2.





**2.5    Retrieval of AOD and *g***
As shown before, maximal sensitivity towards AOD and aerosol phase function is achieved using the
solar azimuth scan. The aim of this study is to develop a simple strategy in order to retrieve AOD and
aerosol phase function, *g* while constraining SSA and SA. A simple method would be to compare the
measurements with the RTM simulations and optimize the aerosol input parameters until we minimize
the differences between measurements and simulations. An iterative approach for a single SRAA scan
would require several hours to finalize. For a typical single day of measurements during the TCAP we
collected at least 3500 spectra using only the azimuth scan. In this context, we believe that a flexible
option is the creation of a look up table (LUT) where the RSP is simulated using geometry related
inputs and numerous aerosol optical properties.

We created the LUT based on different sets of SZA (20˚ to 90˚ in steps of 10˚) and adopted the positive
SRAA as measured by the 2D-MAX-DOAS. The parameters that were fixed are the SSA, 0.98, based
on findings by Müller et al. (2014) and Kassianov et al. (2014) during TCAP. The SA was set to 0.05
representative of the land surface (obtained from the atmospheric transmission by the co-located
MFRSR), and the aerosol extinction height (homogeneous box-height of 1.5 km), though any other
height would give similar results.  We use a typical pressure, temperature, and RH profiles (up to an
altitude of 28 km) measured from radiosondes during TCAP (Berg et al., 2015). Above 28 km the U.S
standard atmosphere was used. The range of parameters that are important and were changed are the
$AOD_{430}$ and *g*. The range of AOD covered was from 0 up to 2.0 AOD in steps of 0.02. The range of *g*
covered was from 0.64 to 0.72 with increments of 0.02. In order to compare with the measurements the
LUT is interpolated to fine grid set points of AOD (in steps of 0.005) and to the average SZA during
the measurements. The AOD and *g* are varied to minimize the following expression:

$$\chi^2 = \sum_{i=1}^{N} \frac{[RSP_M - RSP_{LUT}(AOD, g)]_i^2}{RSP_e^2} \rightarrow min \qquad (1)$$


where $RSP_M$ and $RSP_{LUT}$ are the RSP (arb. units) measured and the simulated in the LUT. $RSP_e$ is the
final estimated RSP error in the measurements (see sect 3.3.1) and *N* represents the number of SRAAs.
A detailed representation of the sensitivity of RSP towards AOD using the SRAA scan and several SZA
is shown in the supplemental information (see Fig. S4).



**2.6 Additional measurements**

The co-located MFRSR (Harrison et al., 1994) and the CIMEL sun photometer (Holben et al., 1998) complement our AOD observations at the TCAP ground site (see details in Table 1). The MFRSR measures total and diffuse solar irradiances at several channels to infer the direct solar radiation component (time resolution of 1 min) while the Sun Sky photometer instrument measures the direct solar beam (time resolution of about 5 min). While both instruments are radiometrically calibrated and work under different principles, a common feature is that they both use the direct sun transmission to derive AODs (Holben et al., 1998; Harrison et al., 1994). The $AOD_{430}$ was calculated using the extinction Angstrom exponent between the standard spectral bands of each instrument (see Table 1). The second generation airborne High Spectral Resolution Lidar (HSRL-2), an improved version of the HSRL-1 (Hair et al., 2008), was deployed aboard the NASA Langley Research Center B200 King Air airplane. HSRL-2 measures particle backscatter coefficients at 355, 532, and 1064 nm, and particle extinction coefficients at 355 and 532 nm (Müller et al., 2014). Similar as the sun photometer, the AOD at 430 nm was calculated using the extinction Angstrom exponent between the standard wavelengths of 355 and 532 nm. Atmospheric temperature and pressure profiles were provided by local radiosondes, which were launched four times a day at the ground site (~ 00, 05, 17, and 23 UTC). The measurement vertical resolution of the sondes was about 10 m reaching a maximum altitude of about 28 km. For this study, the closest radiosonde in time (17 UTC or 13:00 local standard time, LST = UTC-4) is used to prescribe the temperature, pressure and relative humidity in the RTM.

**3. Results and discussions**

**3.1 Effect of the reference spectrum**

The direct sun geometry contains a small amount of RRS light, and is hence not free of RSP contribution. In order to assess the RSP contribution in the direct sun spectra we use two different approaches: (1) a Langley plot type method, where the dRSP obtained with direct sun spectra as reference spectrum is plotted as a function of the SZA, and (2) by interpolating the dRSP measured with small SRAA to the 0˚ (direct sun view). Fig. 6 shows the linear correlation analysis between the direct sun dRSP (binned by SZA) measured on 22 July 2012, low $AOD_{430}$ case (< 0.13), and the air mass factor (AMF), $AMF = 1/\cos(SZA)$. Several direct sun measurements were carried out between SZA of 22˚ (AMF =1.06) and 47˚ (AMF < 1.47) and only one at 78˚ (AMF = 4.8). In order to





quantitatively estimate the RSP in the reference we use the linear correlation analysis applied for SZA
smaller than 50˚ ($R^2 = 0.98$) (see inset plot in Fig. 6). The extrapolation to AMF = 0 yields the absolute
value of the RSP contained in the reference spectrum (RSP value if there were no atmosphere), which is
determined as 0.0053 ± 0.0007 by this method. The value of dRSP at high SZA (78˚) is not considered
here, since there is only one data point and the magnitude is significantly larger (likely due to
atmospheric changes and increasingly distant air masses). To estimate the RSP contained in the
reference with the second method we have analyzed closely the RSP measurements using the solar
azimuth scan for SZA < 45˚ and SRAA close to the sun (SRAA < 40˚). The RSP decreases linearly for
angles close to the sun and an interpolation to 0˚ SRAA yields an RSP value of 0.0035 ± 0.0005. We
use the average of the two methods (0.0044 ± 0.0012) and add this offset to the measured dRSP to
calculate the absolute RSP for comparison with RTM. For assessment of the RSP error, we propagate
the 2-sigma standard deviation (0.0024) in the final uncertainty of $RSP_e$ and in error of AOD and $g$
products (see section 3.3.1).

### 3.1.1 Comparing direct-sun and zenith reference spectrum

To assess the effect of the reference spectrum in the DOAS analysis of the dRSP we compare the dRSP
results using the zenith and direct sun spectra as references; both spectra were recorded at SZA of 28˚.
Figure S5 shows the linear correlation of the dRSP analysis using each reference to analyze all spectra
recorded for azimuth scans for SZA smaller than 70˚ on 22 July 2012. We find a strong linear
correlation ($R^2 ≥ 0.99$) and a slope close to unity (1.023 ± 0.001). The negative offset corresponds to
1.9 % RSP contained in the zenith reference relative to the direct sun. Wagner et al. (2009b) estimated
an RSP of 5 ± 1% in the UV (350 nm) in the noon zenith sky reference by means of RTM simulations
using an AOD of 0.1 measured by a co-located instrument. The strength of the RSP depends on several
factors such as wavelength, the atmospheric conditions (aerosol and cloud optical properties), and the
geometry of measurements. The dRSP in Fig. S5 is color coded by SRAA. The strong SRAA
dependency reflects the sensitivity of RSP to atmospheric scattering processes. The dRSP decreases for
angles close to the sun and increases for larger SRAA. When using the zenith sky as reference the dRSP
obtained would be negative for SRAA < 50˚ and there would be a general negative bias of 1.9 %. Of all
possible viewing directions accessible with ground based measurements the direct sun observation is
the least affected by RRS. In addition, direct sun observations measured with the same instrument
ensure that the spectral resolution and sampling used in the DOAS analysis of all spectra are the same.

### 3.1.2 Calculating references from high-resolution spectra





In principle, high resolution solar spectra (e.g., Chance and Kurucz, 2011) should provide a viable
alternative to direct-sun measurements as reference spectra to retrieve absolute RSP. Such high
resolution spectra need to be convoluted with the instrument slit function prior to their use as reference
spectrum in the DOAS analysis of RSP. We have tested this approach and used high resolution
literature data as a reference spectrum for the analysis of the azimuth scan spectra (1 s integration time),
and find large fitting residuals (RMS ~0.01), that have a strong effect on the retrieved RSP values,
suggesting that this approach is currently of limited value in practice. The causes are likely due to a
combination of reasons, including imperfect knowledge about the wavelength dependent instrument
line shapes, numerical artifacts and assumptions made during convolution, non-linearity of detectors,
and small differences in wavelength calibration. Notably, measuring the direct-sun reference spectrum
in the same instrument as the scattered light spectra inherently accounts for these factors.
**3.2 Effect of aerosol inhomogeneity**
The RTM simulation of RSP and $R_{norm}$ considers aerosol to be uniformly distributed around the
measurement site. To assess if the air mass probed is inhomogeneous we compare quantitatively the
symmetry of the $I_{norm}$ measurements to the left and right side of the sun's disk. The quantitative analysis
of symmetry is defined by the angular asymmetry factor parameter ($AFP_{I_{norm}}$)

$$AFP_{I_{norm}} = \frac{(I_{norm}^L - I_{norm}^R)}{(I_{norm}^L + I_{norm}^R) \cdot 0.5} \qquad (2)$$


where $I_{norm}^L$ and $I_{norm}^R$ are the left and right side measurements of the $I_{norm}$ (counts s$^{-1}$) obtained with
the almucantar scan. The $AFP_{I_{norm}}$ on 17 and 22 July 2012 are shown in the form of a polar plot in Fig.
S5. Over the past few years, CIMEL sun photometers have used a similar approach as a consistency
check to reject pairs of data that exceed 20 % difference and under uniformity the retrieval inversion of
aerosol microphysical properties is applied (Holben et al., 1998). Both days show $AFP_{I_{norm}} < 10$ %,
indicating a high degree of symmetry. In general, the random noise in $AFP_{I_{norm}}$ is on the order of
0.25%. If the $AFP_{I_{norm}}$ shows consistent positive and/or negative values among several SRRAs this
may indicate aerosol inhomogeneity. For example, the increase in AOD at ~ 12:00 LST on 17 July was
accompanied by an average $AFP_{I_{norm}}$ of +2.7 % for the corresponding solar azimuth scan, and
maximum $AFP_{I_{norm}}$ of +10 % at 105° < SRAA < 145°, indicating higher AOD in the south westerly
direction. Ortega et al. (2015b) examined the aerosol extinction inhomogeneity using HSRL-2 data



from overpasses above the TCAP ground site, and found that the AOD varied by about 10 % across the
site at ~13:00 LST on 17 July 2012. By contrast, on 22 July there were no significant differences
visible in the HSRL-2 data, and the symmetry remains all day with average $AFP_{I_{norm}}$ of 0.19 %, and a
standard deviation of 3.3 %.

**3.3 Uncertainty of RSP retrievals of AOD and g**
**3.3.1    RSP retrieval of AOD: maximal sensitivity at low AOD**
Figure 7 shows the simulated RSP in the solar azimuth scan as a function of AOD. The highest
response in RSP to changes in AOD is observed at low AOD, i.e., under conditions when Rayleigh
scattering extinction dominates over aerosol extinction. The sensitivity is highest for small SRAA. A
change of 0.01 AOD when molecular scattering dominates ($AOD_{430} < 0.1$) yields a considerable
decrease in the RSP ($\Delta RSP = 0.004$) for SRAA < 35˚. This change is significantly greater than the
DOAS fit error of 0.0018 presented in section 2.3. The sensitivity decreases for SRAA > 35˚ but still
up to a SRAA of 70˚ the same change in AOD yields a significant (measurable) RSP response. On the
other hand, the sensitivity towards changes in AOD is weaker for AOD greater than 0.3, especially for
low SZA, and for small SRAAs. This is likely due to the dominance of aerosol scattering and few
molecular scattering events. While the reduced sensitivity can in principle be circumvented by
evaluating larger SRAA, such analysis puts more stringent criteria on aerosol homogeneity. The
absolute error in the AOD for any particular SRAA ($AOD_e^i$) was calculated as:

$$AOD_e^i = \left(\frac{RSP_e}{RSP^i}\right) \cdot AOD \qquad (3)$$


where $RSP^i$ is the RSP in the $i^{th}$ SRAA and $RSP_e$ is calculated as the error in the RSP propagated from
the DOAS measured RSP error (~0.0018) and the error in the estimation of the RSP in the reference
(0.0024). Assuming the errors of the measurement to be additive, the final $RSP_e$ is about 0.0028.
Equation 3 is applied to all SRAAs and set of AODs from Fig. 7A. Figure 7B shows the calculated
absolute error in AOD ($AOD_e$) using all elements from Fig. 7A and weighted as follows:

$$AOD_e = \frac{\Sigma \left(\frac{AOD_e^i}{RSP_e}\right)^2}{\Sigma \left(\frac{1}{RSP_e}\right)^2} \qquad (4)$$




The weighted RSP relative error $\left(\frac{RSP_e}{RSP^i}\right)$ following the same approach is also shown in percentage in
Fig. 7B. Under high AOD conditions ($\sim$ 0.4) the absolute $AOD_e$ is 0.02. The $AOD_e$ decreases
significantly for AOD $\leq$ 0.1, with uncertainties of about 0.004 at AOD of 0.1, and 0.0025 at AOD of
0.05. As mentioned before the information content on aerosols using the solar azimuth scan is
enhanced at large SZA when RSP values are larger. At SZA = 70˚ (Fig. 7C), the errors decrease further
for low AOD as is illustrated in Figure 7D. The $AOD_e$ is 0.014, 0.003, and 0.002 for an AOD of 0.4,
0.1, and 0.05, respectively. The error scales roughly with $\cos(SZA)$, indicating that the highest
sensitivity of RSP based AOD retrievals is at high SZA and low AOD.

**3.3.2    Aerosol phase function**
As shown in Figure 5, the phase function parameter $g$ has the strongest effect on the simulated RSP for
small SRAA. A decrease of the $g$ parameter, i.e., decrease of aerosol forward scattering probability,
leads to an increase of the RSP due to a higher contribution of molecular scattering at this direction.
The radiances also show a significant sensitivity towards $g$ for small SRAA, as previously shown by
Frieß et al. (2006). For a fixed AOD and a change in $g$ of $\pm$ 0.04 the RSP difference is about 0.03 for a
SRAA of 5˚, which is two times greater than the RSP error. In general, measurements at small SRAA
carry most information, and are highly recommended (Holben et al., 1998; Frieß et al., 2006). In
addition, the quality of the retrieval of $g$ is expected to improve for high SZAs when there is an
increase in the information content of the scattering angle coverage (Torres et al., 2013; Dubovik et al.,

448    2000).


**3.4    Comparison of measurements and simulations**
We compare simulated and measured RSP for several SRAA ranges in Fig 8. The example shown in
Fig. 8 is obtained by applying the retrieval approach explained in section 2.5 for the solar azimuth scan
(SZA = 66.5˚) on 22 July 2012. Four sets of SRAAs are used: (A) 5˚ to 20˚, (B) 5˚ to 60˚, (C) 5˚ to
120˚, and (D) 5˚ to 170˚. Three values for $g$ are used to show the sensitivity towards the phase function.
The AOD retrieved with each $g$ is shown in the label box. The residuals, defined as the difference
between measured and simulated RSP (minimizing equation 1) are shown in the bottom panel below
each comparison. The gray shaded area (behind the residuals) represents the RSP error ($\pm$ 0.0028)
defined before. The computed RMS errors (RMSE) are also shown. The comparison of the RSP
constrained by few SRAAs (< 20˚, Fig. 8A) show that all the residuals lie within the error bars





independently of $g$. However, the variability of the retrieved AOD is significant for each $g$ and
maximum $\Delta$AOD of 0.025 is obtained. When using more SRAAs (Figures 8B, 8C, 8D) the spread in
AOD values is reduced.  The maximum $\Delta$AOD obtained with using either 5˚ to 60˚, 5˚ to 120˚ or 5˚ to
170˚ SRAAs is 0.010. Significant residuals (greater than the RSP error) are obtained for $g$ larger than
0.68. The residuals obtained with the $g$ of 0.64 are always within the error bars of the measured RSP
indicating that this $g$ (for SRAA < 40˚) is in excellent agreement with the $g$ of 0.65 reported by the
CIMEL sun photometer close to this time. This further suggests that SRAA close to the sun are needed
and essential in order to maximize the sensitivity of the aerosol phase functions.

**3.5    Optimized observing strategy**
We have optimized a retrieval strategy such that at high SZA (> 50˚) we use SRAA in the range of 5˚ to
60˚, and for smaller SZA we use the full azimuth scan (5˚ < SRAA < 170˚). This was motivated by the
fact that for 5˚ < SRRA < 60˚ the AOD and $g$ are stable, and show the minimal RMSE and maximal
information content at high SZA for this range. On the other hand less information content is achieved
at low SZA and more SRAA are needed. This optimization may be important in the presence of broken
clouds. In this case, as long as there is homogeneity for SRAA < 60˚ the retrieval strategy presented
here may yield good results.
TCAP represented the first deployment of the CU 2D-MAX-DOAS instrument. The geometry of
measurements was motivated by retrieving simultaneously trace gas and aerosol extinction profiles by
means of the EA scan and by testing the solar azimuth scan for the first time (table 2). The acquisition
time of the solar azimuth scan was 2 min. However the time resolution of the retrieved products is
about 20 min due to the 85 % duty cycle of the EA scan in the single repetition of both EA and solar
azimuth scan. The fast mode of measurements in the almucantar limits the retrieval of many typical
DOAS species such as the oxygen dimer ($O_2$-$O_2$, or $O_4$) and other trace gases (such as $NO_2$, HCHO,
CHOCHO, etc). Incrementing the time resolution in the solar azimuth scan would mean that $O_4$ could
be measured and have an additional piece of information in the retrieval of aerosol
optical/microphysical properties. Future deployments may consider a combination of SRAA scans
with longer integration time to also obtain trace gases, and EA scans for a subset of SRAA to obtain
trace gas vertical profiles. In addition, future deployments with 2D capabilities might consider the
solar principal plane sky geometry, which is similar to the almucantar scan but in the principal plane of
the sun (see Holben et al., 1998). This geometry would be very similar to the typical off-axis scan, i.e.,
high sensitivity towards the lower part of the atmosphere. In addition SRAA would be measured





giving information about phase functions. Furthermore, a future deployment may dedicate a full day to
direct sun observations in order to apply the Langley plot to more SZA, ideally during constant diurnal
AOD conditions such as in Mauna Loa, HI.

**3.6    Comparison with CIMEL sun photometer, MFRSR, and HSRL-2**
Figure 9 compares the diurnal variability of $AOD_{430}$ and $g$ with independent measurements by
MFRSR, CIMEL sun photometer, and HSRL-2 instruments for (A) 22 July (low AOD case) and (B) 17
July (high AOD case) 2012. The molecular scattering optical depth represented with the discontinuous
gray line is calculated with the method reported by Bodhaine et al. (1999) using the temperature and
pressure profiles from the local radiosonde (launched at 13:00 LST). Considering a diurnal direct sun
geometry the molecular scattering optical depth is weighted by the air mass factor $1/\cos(SZA)$. On 22
July the retrieved aerosol $AOD_{430}$ is below the molecular scattering regime all of the day for MFRSR,
DOAS and HSRL-2, and most of the time for the CIMEL sun photometer. Under these conditions the
uncertainties of the AOD retrieved from the solar beam extinction approach, i.e., MFRSR and CIMEL
sun photometer, might be greater than 0.01 AOD, which is a typical error after calibration (Holben et
al., 1998; Harrison et al., 1994). We have adopted this ideal error of 0.010 AOD for the MFRSR and
CIMEL sun photometer in Fig. 9A. The error bars of the 2D-MAX-DOAS are those discussed in
section 3.3.1. In general, the errors are smaller at high SZA, as discussed in Sect. 3.3.1, and the largest
errors are ~0.012 at noon. The comparison of the $AOD_{430}$ retrieved by DOAS compares well, and is
generally within the combined error bars with the other measurements. The comparison is best in the
morning, and DOAS agrees better with the MFRSR throughout the day; there is only marginal
agreement with the CIMEL sun photometer in the afternoon. At noon, there is a small increase in
$AOD_{430}$ of about 0.05 and the response of this change is greater for the 2D-MAX-DOAS than for the
MFRSR and Sun photometer likely due to maximum sensitivity to small changes in AOD. A power
outage inside the seatainer restricted measurements after 17:30 LST.

The average diurnal difference of $AOD_{430}$ (relative to the 2D-MAX-DOAS) on 22 July is +0.0199 ±
0.014 (CIMEL), +0.003 ± 0.019 (MFRSR), and -0.011 ± 0.014 during the overpass of the HSRL-2.
The $AOD_{430}$ measured by HSRL-2 during two overpasses is slightly lower than the $AOD_{430}$ measured
by the CIMEL sun photometer, agrees closely in one instance with the 2D-MAX-DOAS, and the
closest agreement is observed for MFRSR. Note that the HSRL AOD values correspond to the layer
between the surface and about 7 km. In general, 90 to 95 % of the aerosol extinction is estimated to be





below the ~ 7 km. The average diurnal difference in $AOD_{430}$ shows that the CIMEL sun photometer is
0.017 greater than MFRSR. In general a good agreement is reflected in the linear correlation analysis
between the 2D-MAX-DOAS and CIMEL sun photometer: $DOAS_{AOD}$ = -(0.013 ± 0.010) + (0.96 ±
0.09)·$CIMEL_{AOD}$ ($R^2$ = 0.82); between 2D-MAX-DOAS and MFRSR: $DOAS_{AOD}$ = -(0.029 ± 0.020) +
(1.32 ± 0.21)·$MFRSR_{AOD}$ ($R^2$ = 0.64); and between CIMEL sun photometer and MFRSR: $CIMEL_{AOD}$
= -(0.028 ± 0.009) + (1.45 ± 0.10)·$MFRSR_{AOD}$ ($R^2$ = 0.91). Notably, the offset is larger than 0.02 in
some instances, highlighting the importance of instrument comparisons under low AOD conditions.

On 17 July the $AOD_{430}$ reached values of 0.6 at noon (Fig. 9B). The high AOD and low RSP signal
from 11:00 to 14:00 LST limited the retrieval of AOD and $g$ from the 2D-MAX-DOAS. As shown in
Fig. 7 the RSP decreases significantly at high AOD and low SZA likely due to dominance of multiple
aerosol forward scattering. On this day averaged RSP values obtained with the solar azimuth scans
from ~11:00 to 14:00 were similar to the RSP error limiting the analysis. This could be circumvented
in the future by conducting almucantar scans at a lower EA. Under conditions when a retrieval is
warranted, the comparison of the difference in $AOD_{430}$ (relative to the 2D-MAX-DOAS) is -0.027 ±
0.03 (CIMEL), +0.005 ± 0.027 (MFRSR). The AOD from CIMEL sun photometer is 0.035 ± 0.015
greater than that from MFRSR. The linear correlation between the 2D-MAX-DOAS and CIMEL sun
photometer is: $DOAS_{AOD}$ = -(0.017 ± 0.034) + (0.95 ± 0.08) x $CIMEL_{AOD}$ ($R^2$ = 0.88); between 2D-
MAX-DOAS and MFRSR: $DOAS_{AOD}$ = -(0.025 ± 0.027) + (1.00 ± 0.07) x $MFRSR_{AOD}$ ($R^2$ = 0.91);
and between CIMEL sun photometer and MFRSR: $CIMEL_{AOD}$ = -(0.020 ± 0.015) + (1.03 ± 0.04) x
$MFRSR_{AOD}$ ($R^2$ = 0.97).

Clearly under high AOD conditions the maximum AOD diurnal difference of ~ 0.027 accounts for less
than 5 % of the AOD. On the other hand, the diurnal differences between instruments under low AOD
account (~0.02) for about 20 % of the AOD. Overall good agreement is reflected in the linear
regression analysis of pooled data from both case study days: $DOAS_{AOD}$ = -(0.019 ± 0.006) + (1.03 ±
0.02)·$CIMEL_{AOD}$ ($R^2$ = 0.98), $DOAS_{AOD}$ = -(0.006 ± 0.005) + (1.08 ± 0.02)·$MFRSR_{AOD}$ ($R^2$ = 0.98),
and $CIMEL_{AOD}$ = (0.013 ± 0.004) + (1.05 ± 0.01)·$MFRSR_{AOD}$ ($R^2$ = 0.99).

**3.7    Context with literature: advantages and limitations**
According to Holben et al. (1998) the AOD uncertainty of newly calibrated Sun photometers is ± 0.01
for typical visible wavelengths, and ± 0.02 for shorter wavelengths. In particular, the error in AOD



becomes highly sensitive to the calibration error at low AOD. For example, for $AOD_{440} < 0.05$ and 5%
calibration error the AOD uncertainty can reach 44 %. On the other hand, the error in AOD decreases
dramatically for the same calibration error if solar scattering measurements are used (Holben et al.
1998). Our innovative retrieval strategy for AOD and $g$ is based on solar scattered light, but
circumvents the calibration uncertainty outlined above, and provides robust measurements under low
AOD conditions. Our measurements are inherently calibrated, i.e., do not require radiance calibration
which is subject to drift, and needs frequent sensor attention during field operation. The RSP-based
retrievals only rely on relative radiance measurements in the SRAA and hyperspectral domain, which
makes them particularly useful for long-term observations in remote environments.
The diurnal error in AOD of direct sun transmission measurements is also subject to the optical path
through the Earth's atmosphere. In general, the nominal error in AOD will change with the air mass
factor (cos(SZA)) and potentially needs to be scaled accordingly leaving smaller errors at high SZA
(Sinyuk et al., 2012). One important advantage of RSP-based retrievals is that the aerosol information
content is enhanced at low AOD. RSP constraints to column aerosol optical properties are
complementary to $O_2$-$O_2$ measurements that are widely used to infer information about clouds and
aerosols (Baidar et al., 2013; Gielen et al., 2014; Wagner et al., 2014; Ortega et al., 2015b; Volkamer et
al., 2015). The synergistic use of RSP and $O_2$-$O_2$ holds great potential to better assess profile and
column properties of aerosols and clouds, and currently remains largely unexplored.
**4.    Summary and conclusions**
In this work we present a detailed analysis of RRS using direct sun and solar almucantar
measurements of scattered solar photons by the CU 2D-MAX-DOAS instrument (see part 1, Ortega et
al., 2015a). The rapid solar azimuth scan, i.e., integration time of 1 s and total acquisition time of ~2
min to measure from -180˚ to 180˚ SRAA in steps of 5˚ relative to the sun, provide robust means to
simultaneously retrieve AOD and the column integrated aerosol phase function (simplified by the
asymmetry parameter, $g$). We conclude the following:
•   Measurements of RSP have maximum sensitivity towards retrieving AOD and $g$ under

582       molecular scattering conditions. This is demonstrated with RTM simulations of the RSP using

583       diurnal solar azimuth geometry. The highest sensitivity towards both $g$ and AOD is achieved if

584       using small SRAA ($\leq 5˚$).

•   The error in the RSP based retrieval of AOD and $g$ is limited by the uncertainty about RSP





contained in the reference spectrum. We minimize the error by retrieving near-absolute RSP
using a direct sun reference spectrum recorded with the same instrument. The direct sun
spectrum is also affected by RRS. We estimate $RSP_{DS} = 0.44 \pm 0.22$ %, compared to $RSP_{zenith} =$
$2.34 \pm 0.22$ % (SZA = 28˚, $AOD_{430} = 0.11$). Direct sun observations at low SZA systematically
minimize RSP, and are most valuable for precise AOD and $g$ retrievals.
•   RSP based retrievals of AOD and $g$ have higher sensitivity at high SZAs, and low AOD. This is
complementary to existing techniques that operate on solar transmission. The absolute error are
about 0.02, 0.004, 0.0025 and 0.0005 for $AOD_{430}$ of 0.4, 0.1, 0.05 and 0.01, respectively (4-
5 % relative error), at SZA = 35˚. The errors decrease with increasing SZA, and absolute errors
are 0.014, 0.003, 0.002, and 0.0004 for $AOD_{430}$ of 0.4, 0.1, 0.05 and 0.01, respectively (3-4 %
relative error), at SZA = 70˚.
•   Retrievals based on RSP measurements at a subset of SRAA hold potential to measure AOD
under broken cloud conditions. Clear sky and broken cloud conditions can be identified using
the color index, and AOD retrievals under such conditions warrant further study.
•   The RSP retrieval of AOD and $g$ consist is inherently calibrated, since it relies only on relative
intensity changes that are measured in the hyperspectral domain, and at various SRAA.
Combined with the high sensitivity at low AOD and high SZA makes measurements of RSP
particularly useful to conduct long-term time series measurements in remote environments,
such as the arctic, or remote ocean environments at tropical latitudes.
•   The retrieval strategies may be optimized by conducting azimuth scans with longer integration
time, at solar EA and lower EAs, and by conducting EA scans for a larger subset of SRAA to
simultaneously measure azimuth distributions and vertical profiles of trace gases. 2D-MAX-
DOAS measurements in the solar principal plane sky geometry (similar to the almucantar scan)
would further increase the sensitivity towards the lower part of the atmosphere.
*Acknowledgements*. The instrument was developed with support from the NSF-CAREER award ATM-
0847793; US Department of Energy (DOE) award DE-SC0006080 supported the TCAP deployment
(RV). Ivan Ortega is recipient of a NASA Earth Science graduate fellowship. Larry Berg is supported
by the DOE Atmospheric System Research (ASR) Program. The Pacific Northwest National
Laboratory is operated by Battelle Memorial Institute under contract DE-AC06-76RLO 1830. The
CIMEL sun photometer data were collected by the U.S. Department of Energy as part of the
Atmospheric Radiation Measurement Program Climate Research Facility (ARM) and processed by the
National Aeronautics and Space Administration's Aerosol Robotic Network (AERONET). Support for




the HSRL-2 flight operations during TCAP was provided by the DOE ARM program: Interagency
Agreement DE-SC0006730. We are grateful to Tim Deutschmann for providing support with the
McArtim RTM. We thank Caroline Fayt and Michel van Roozendael for providing the WinDOAS
software.

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



**Tables:**

Table 1. Suite of measurements and products used in this work.

| Ground based instruments | | | | |
|---|---|---|---|---|
| **Instrument** | **Principle of measurement** | **Absolute radiometric calibration (YES/NO)** | **Products** | **Reference** |
| 2D-MAX-DOAS | Solar scattered light | NO | AOD and $g$ (430 nm) | Ortega et al. (2015a) |
| MFRSR | Total and diffuse solar irradiances | YES | AOD (430 nm) calculated using the Angstrom exponent between the standard wavelengths of 415 and 500 nm. | Harrison et al. (1994) |
| CIMEL sun photometer | Direct solar beam and diffused sky radiation | YES | Level 2.0: AOD (430 nm) calculated using the Angstrom exponent between the standard wavelengths of 340 and 440 nm and $g$ (440 nm) | Holben et al. (1998) |
| Radiosondes | Weather balloon that measures various atmospheric parameters | N/A | Vertical profiles of temperature, pressure and humidity (4 times/day) | Berg et al. (2015) |
| Airborne instruments | | | | |
| HSRL-2 | backscatter and extinction coefficients | | AOD (430 nm) - calculated using the Angstrom exponent between the standard wavelengths of 355 and 532 nm. | Muller et al. (2014) |


Table 2. Geometry of measurements and configuration used during the TCAP field campaign.

| Mode | EA (˚) | AA (˚) | SRAA (˚) | Total acquisition Time (min) |
|---|---|---|---|---|
| 1 | 1,3,6,8,10,30,45,90 | 0, 180 | variable | 16 - 17[a] |
| 2 | Solar elevation (90 – $\theta_0$) and 45 | variable | 5, 10, 15. . . 180 (left and right) | 2 - 3[b] |
| 3 | Solar elevation | variable | 0 | 2 – 6 |

[a]Integration time of 60s at each $\alpha$. [b]Integration time of 1s at each SRAA.



**Figures:**

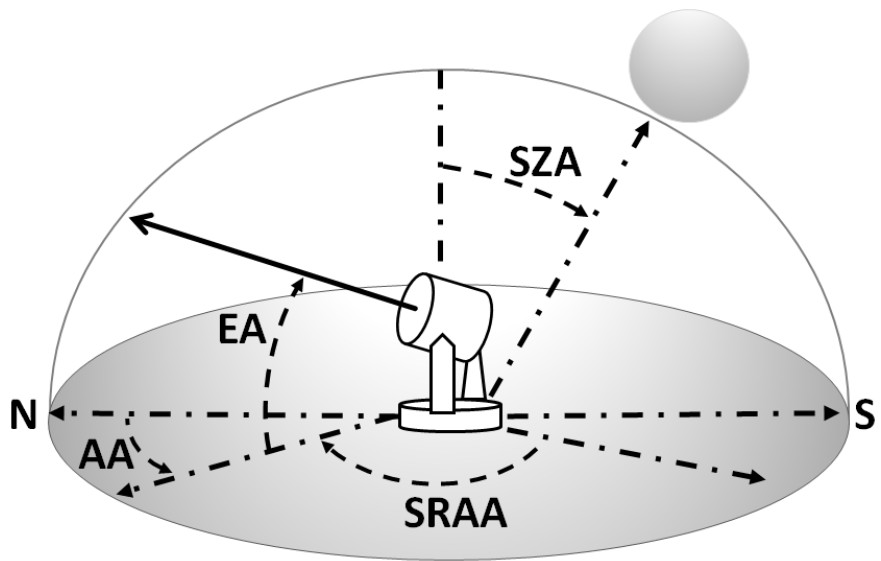


Fig. 1. Sketch of measurement geometry used with the 2D-MAX-DOAS. The solid line coming out
from the telescope represents the azimuth angle (AA) with respect to North characterized by the
elevation angle (EA) and solar relative azimuth angle (SRAA). SZA is the solar zenith angle.

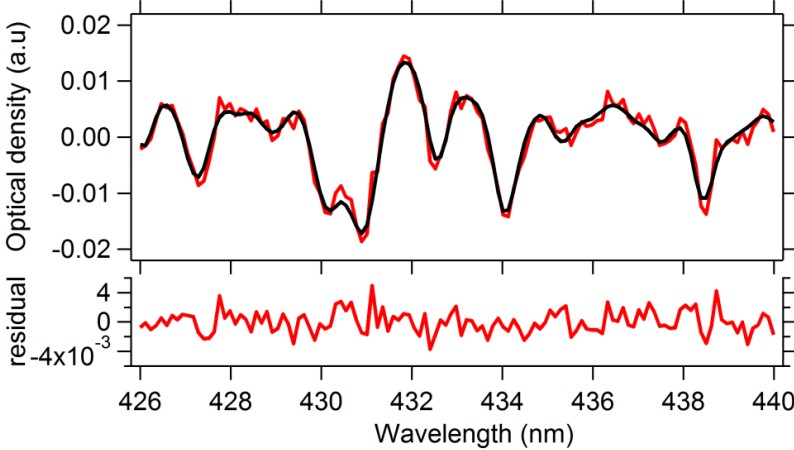


Fig. 2. Top: example of spectral proof for the detection of dRSP (1 s integration time) using the solar
azimuth scan on 22 July 2012 at 7:43 LST (SZA = 66.3°, SRAA = 120°, EA = solar EA). The red line
represents the measured spectra and black line is the fitted normalized Ring cross section. The dRSP is
$0.0502 \pm 0.0011$. Bottom: residual from the DOAS fit, $RMS_{meas} = 1.58 \times 10^{-3}$, is in good agreement with
the shot-noise $RMS_{theo} = 1.40 \times 10^{-3}$ based on photon counting statistics.





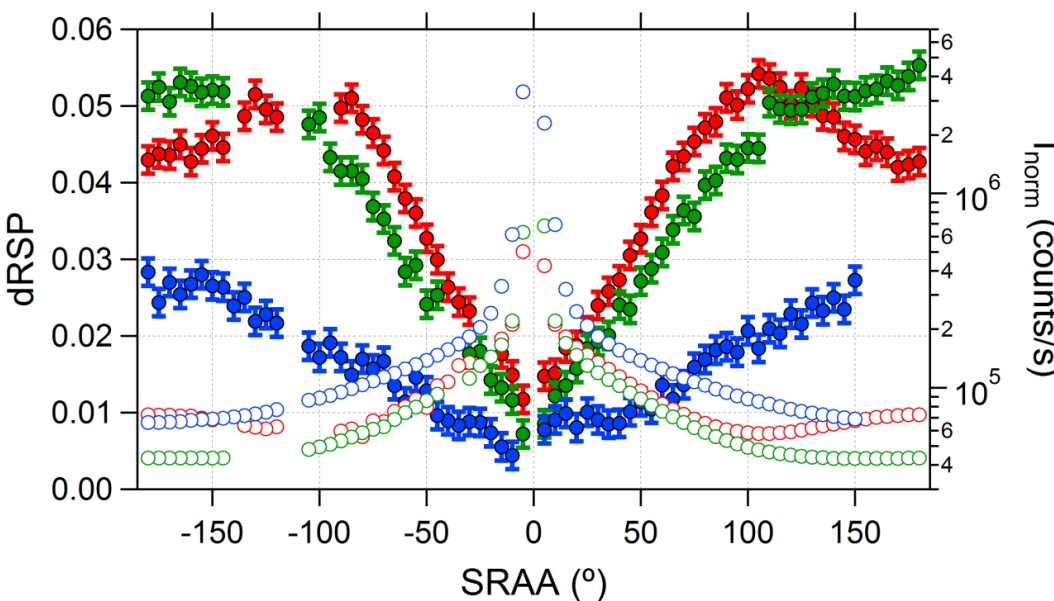

Fig 3. Example of the SRAA dependence of dRSP (filled circles) and $I_{norm}$ (open circles) measured at
the solar elevation near 430 nm for three SZAs: (red) 66.5˚ (7:42 LST), (green) 49.0˚ (9:17 LST), and
(blue) 22.0˚ (12:40 LST).



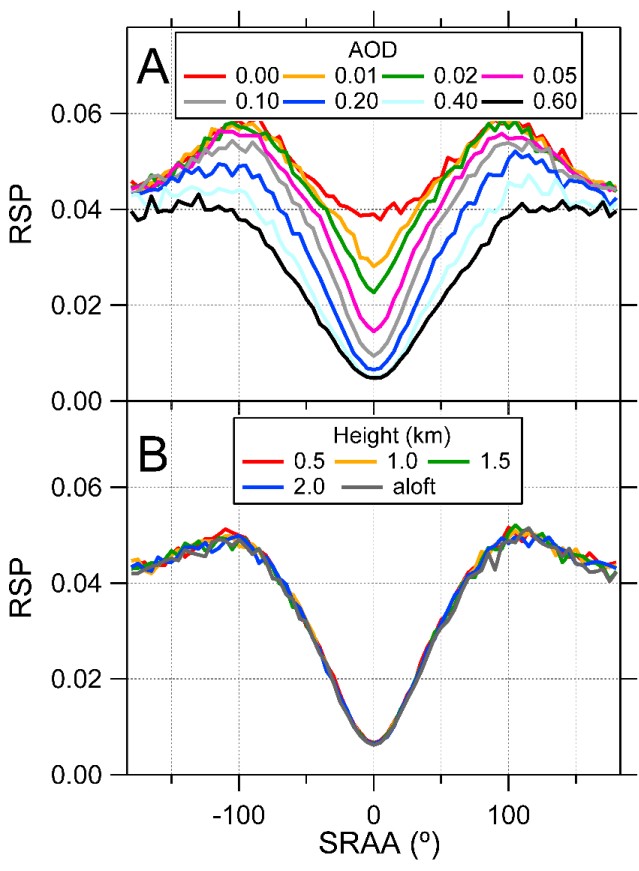


Fig. 4. Sensitivity study showing that simulated RSP (430 nm) is (A) a strong function of AOD, and (B)

insensitive to the aerosol vertical distribution. (A) AOD is varied, keeping aerosols homogeneously

distributed (box profile) up to 1.5 km altitude. (B) The aerosol extinction vertical distribution is varied

for a constant AOD of 0.2. The simulation is for SZA = 70˚, SSA = 0.98, g = 0.70, SA = 0.05.





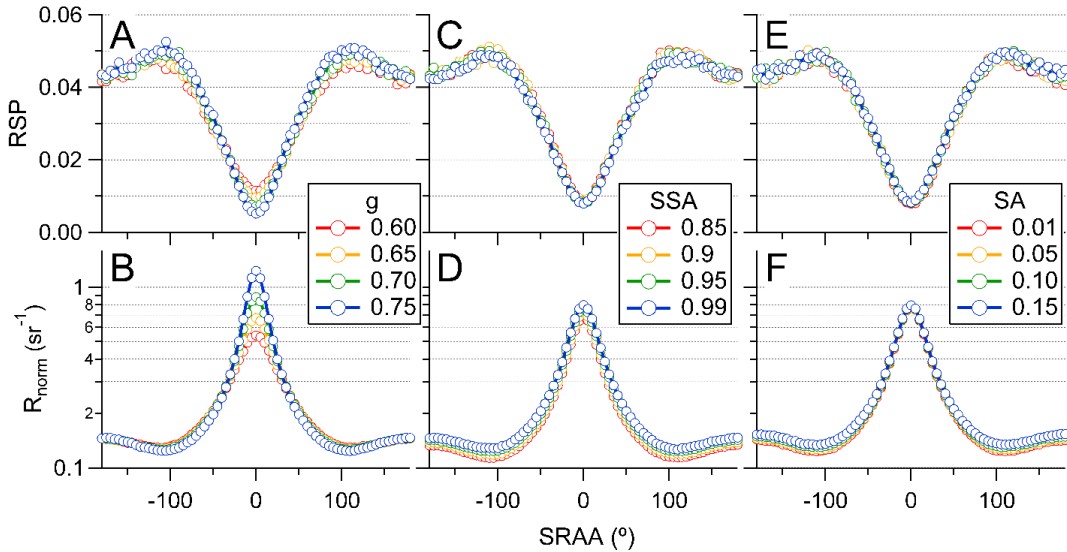

Fig. 5. Sensitivity study showing simulated RSP (top row) and sun-normalized radiances, $R_{norm}$ ($sr^{-1}$)
defined as the ratio of the radiance ($W \cdot m^2 \cdot sr^{-1}$) in the geometry indicated to the solar irradiance ($W \cdot m^2$)
(bottom row) at 430 nm for several input parameters: (A, B) $g$, (C, D) SSA, and (E, F) SA. The
simulations were carried out assuming a box extinction profile (1.5 km height) with an AOD of 0.2, and
SZA = 70˚.

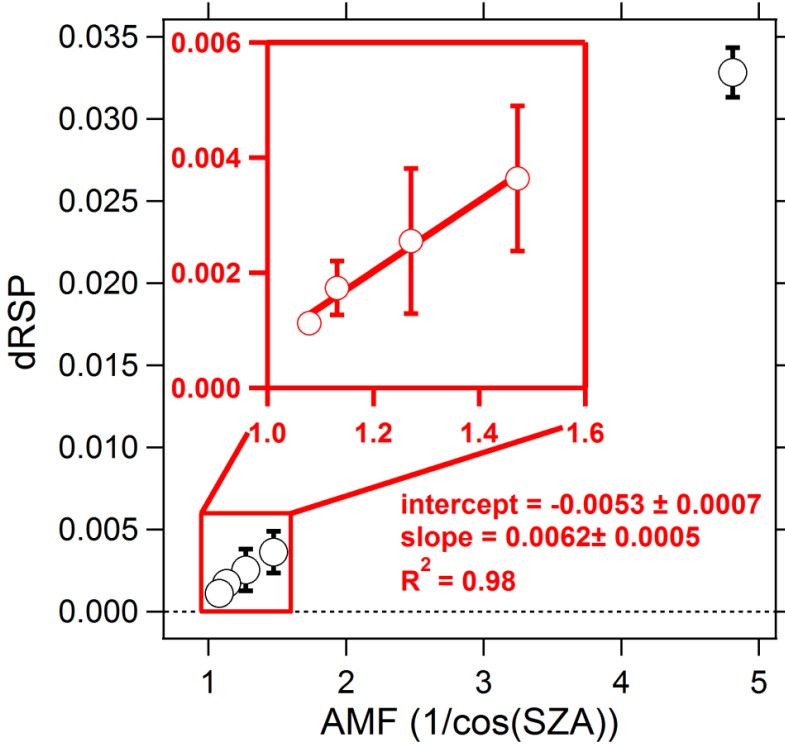


Fig. 6. Direct sun dRSP as a function of the air mass factor, AMF = 1/cos(SZA). All direct sun spectra

measured on 22 July 2012 were evaluated (SZA binned) relative to a direct sun reference spectrum

measured at SZA = 28°. The insert shows the zoom-in of the linear correlation plot used to

quantitatively determine the offset at AMF = 0 (see text for details).

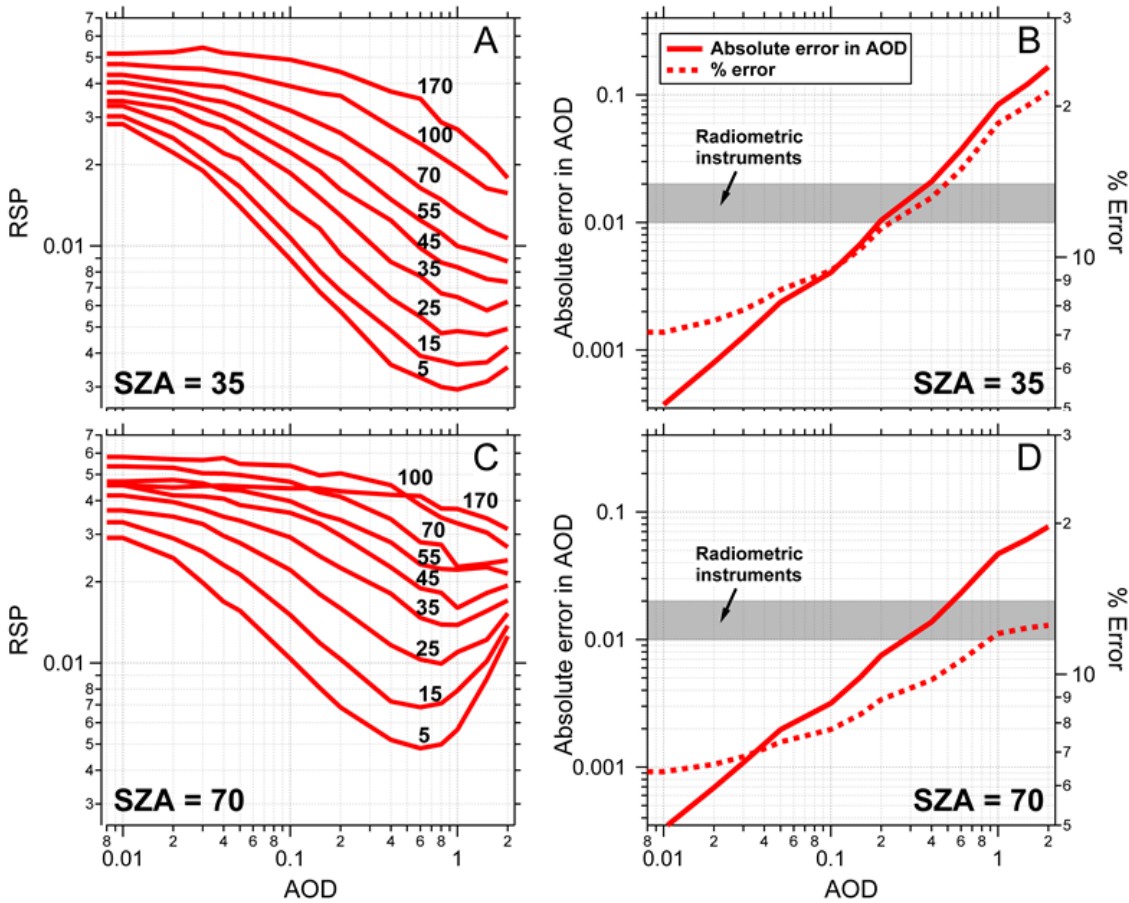


Fig 7. Simulated RSP as a function of AOD for 5° < SRAA < 100° at (A) SZA = 35°, and (C) SZA =
70°. Additional parameters are g = 0.72, SA = 0.05, SSA= 0.98. (B and D) Absolute error in AOD and
percentage error calculated with equation 3 (see text for details). An AOD error range of 0.01 − 0.02 is
indicated with the gray shadow area. A value of 0.01 is typical of newly radiometrically calibrated
instruments.





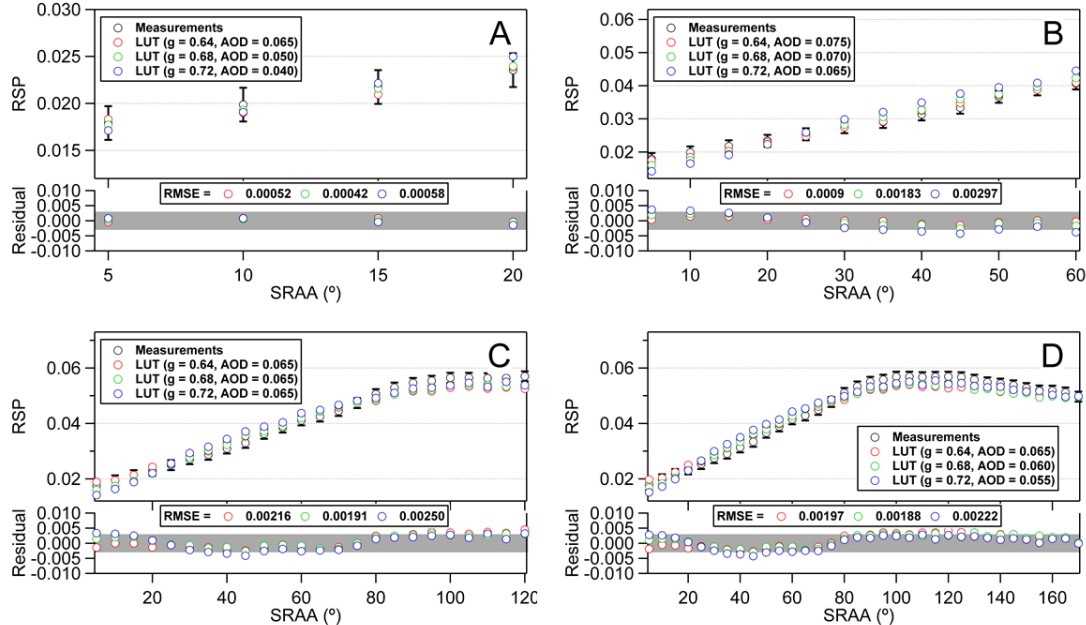


Fig 8. Comparison of measured RSP (black open circles) and simulated RSP (open circles) with the LUT using three different $g$ (red: $g = 0.64$, green: $g = 0.68$, and blue: $g = 0.72$ ). The examples shown here represent the best fit that minimizes equation 1 for different ranges of SRAA: (A) 5 to 20˚, (B) 5 to 60˚, (C) 5 to 120˚, and (D) 5 to 170˚ SRAA. The retrieved AOD in each case is indicated in the labels.






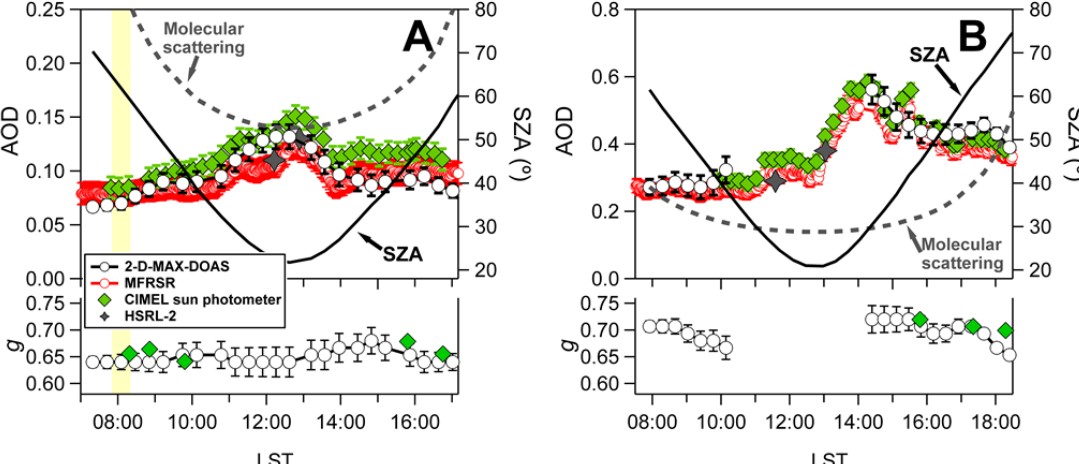


Fig 9. Time series of AOD comparing the 2D-MAX-DOAS with MFRSR, CIMEL sun photometer and
HSRL-2 under (A) low AOD case on 22 July and (B) under high AOD on 17 July. The dashed gray line
represents the molecular optical depth. The retrieved $g$ from 2D-MAX-DOAS (430nm) and CIMEL
(440 nm) are shown in the bottom plot. The yellow shading in A represents the time period used in the
example of Fig. 8.