# Peer review of "The CU 2D-MAX-DOAS instrument - part 2: Raman Scattering Probability Measurements and Retrieval of Aerosol Optical Properties"

_Atmospheric Measurement Techniques, 2015_

## Referee Comment (RC1) · Anonymous Referee #1 · 13 Feb 2016

Review of the paper

'The CU 2D-MAX-DOAS instrument -part 2: Raman Scattering Probability Measurements and 1 Retrieval of Aerosol Optical Properties'

by Ivan Ortega et al.

The paper by Ortega et al. describes a new method for the determination of the aerosol optical depth and information on the aerosol scattering phase function from azimuth scans of spectrally resolved observations of scattered sun light. The method has two important advantages: a) it is based on the so called Ring effect, which can be retrieved with high accuracy without the need of an absolute radiance calibration. b) the

sensitivity of the method is particularly high for small AOD, for which other instruments tend to have large uncertainties.

The new method has the potential to improve AOD observations especially at low AOD. The paper is innovative, and the proposed method is well described and the first results are very promising.

I recommend publication in AMT after two major issues are addressed:

Major comments:

A) The authors describe the aerosol phase function by the Henyey-Greenstein parameterisation. This is a very simple parameterisation based on only one parameter. It is well known that true aerosol phase functions can not be well described by the HG parameterisation. Especially close to the forward direction, the deviations can become rather large. This is the range of scattering angles, which is explored in this study. I recomment that the authors repeat their radiative transfer simulations using more realistic aerosol phase functions (e.g. using Mie phase functions based on the sun photometer measuremens as input). If this is not possible, the authors should demonstrate that the use of a HG parameterisation in their study is justified.

B) A direct sun spectrum is used as Fraunhofer reference spectrum and the corresponding is determined by two different approaches. However, in my opinion, both approaches are based on false assumptions (see below). My suggestion would be to simply derive the RSP from radiative transfer simulations: Here I propose a simply procedure: 1) the radiance and the RSP are calculated for a scattered sun light observations in the direction of the sun. 2) the radiance of the direct sun is calculated for the same direction (the RSP for the direct light is assumed as zero). 3) The effective RSP is caluclated as the average RSP of both contributions (direct and scattered sun light) weighted by their respective radiances.

Reasons, why in my opinion both approaches in the paper are wrong:

On page 11, line 335 both methods are introduced: '(1) a Langley plot type method, where the dRSP obtained with direct sun spectra as reference spectrum is plotted as a function of the SZA, and (2) by interpolating the dRSP measured with small SRAA to the 0 degree (direct sun view).'

Method (1) is based on the assumption that the dRSP is a linear function of 1/cos(SZA). I expect that this is in general not the case.

Method (2) would only be justified if a smooth transition of the RSP between measurements of scattered and direct sun light could be expected. In my opinion this is not the case.

Minor comments:

Page 2, line 38: 'Active steps to minimize RSP in the reference spectrum help to reduce the uncertainty in RSP retrievals of AOD and g.'

I disgaree with this statement: Not the RSP value itself should be minimised, but rather the uncertainty of the RSP value. In the proposed method, RSP from measurements and simulations are compared. In this comparison, the absolute, but not the relative deviation between both data sets is minimised. This means that the uncertainty of the RSP of the reference spectrum should be minimised (but not the value of the RSP itself).

Page 4, line 97: 'The quantitative analysis of RRS by DOAS was introduced by Wagner et al. (2004, 2009a) with the so-called "Raman Scattering Probability...'

I think additional pioneering studies should be mentoned, e.g.:

Langford, A. O., Schofield, R., Daniel, J. S., Portmann, R. W., Melamed, M. L., Miller, H. L., Dutton, E. G., and Solomon, S.: On the variability of the Ring effect in the near ultraviolet: understanding the role of aerosols and multiple scattering, Atmos. Chem. Phys., 7, 575–586, 2007.

de Beek, R., Vountas, M., Rozanov, V. V., Richter, A., and Burrows, J. P.: The Ring effect in the cloudy atmosphere, Geophys. Res. Lett., 28, 721–724, 2001.

Vountas, M., Richter, A., Wittrock, F., and Burrows, J. P.: Inelastic scattering in ocean water and its impact on trace gas retrievals from satellite data, Atmos. Chem. Phys., 3, 1365–1375, 2003.

Page 6, line 154 : 'In this work, we focus only on the almucantar scan at solar EA.'

Why was this procedure chosen? If in addition, also measurements at other elevation angles were used (e.g. the scan at 45° EA) additional information could be obtained. (or the consistency of the results could be checked)

Page 7, line 191: RSP is not measured but retrieved

Page 7, line 204: 'Systematic errors in the retrieval of dRSP were quantified by means of sensitivity studies.'

Did the authors investigate the effect of instrument straylight (or its correction by the analysis software) on the RSP results?

Page 9, line 263: 'The sensitivity studies in Figs 3, 4, and in the supplement confirm that the RSP does not depend on the aerosol vertical distribution.'

This statement is based on measurements at rather high elevation angles. I expect that for measurements at low elevation angles (which are often used for MAX-DOAS observations) the RSP becomes dependent on the aerosol vertical distribution. The authors should discuss this aspect.

Page 14, line 408: 'The highest response in RSP to changes in AOD is observed at low AOD'

I am not sure if this statement is supported by the results shown in Fig. 7. The authors should possibly use linear scales. How exactly is 'response in RSP to changes in AOD' defined

Page 19, line 585: 'The error in the RSP based retrieval of AOD and g is limited by the uncertainty about RSP contained in the reference spectrum.'

I think this statement is not supported by the shown results. In particular, I think the choice of a direct sunlight Fraunhofer reference spectrum is probably not the best choice, because the instrument response (in particular instrument stray light) is probably different for scattered and direct light observations. My suggestion would be to use a Fraunhofer reference spectrum of scattered sun light from the same measurement sequence (measured in zenith direction) and to simulate instead of the RSP the dRSP.

---

## Referee Comment (RC2) · Anonymous Referee #2 · 22 Feb 2016

Ortega et al., describe derivation of aerosol optical depth and Henyey-Greenstein assymetry factor from solar almucantar measurements (-170° to 170°, 5° steps) of Rotational Roman Scattering probability. The method applies DOAS technique to hyperspectral intensity measurements to derive differential RSP in 426 – 440 nm window and therefore does not require absolute radiometric calibration. The authors present radiative transfer simulations at 430 nm using Monte Carlo model to demonstrate sensitivity of the RSP to AOD, H-G asymmetry factor, aerosol profile, relative solar azimuth angle, and solar zenith angle. They conclude that RSP is independent of the aerosol profile and has low dependence on single scattering albedo and surface reflectivity. On the other hand, RSP has high sensitivity to total AOD and H-G asymmetry factor

especially at small RSAA, small AOD and large SZA. Based on these simulations they develop a method to minimize difference between retrieved (426 – 440 nm) and simulated RSP at 430 nm. Direct sun spectrum is used as a reference Fraunhofer spectrum to minimize amount of RPS in the reference spectrum. RSP in the reference spectrum is derived from Langley plot analysis of the zenith and direct sun spectra. The method is applied to 2 days, one with low and one with high AOD, during TCAP filed campaign (1 July – 13 August 2012). The retrieved AOD are compared to co-located measurements by CIMEL, MFRSR, and HSRL-2. Reasonable agreement in diurnal variability is achieved between CU 2D-MAX-DOAS, CIMEL, and MFRSR. The method is well described and the paper is well organized. I recommend publishing the paper after some modifications.

Major comments:

1. One of the main assumptions of the method is that solar almucantar measurements of RSP are independent of aerosol profiles based on the simulations at SZA 35° and 70°. This might not hold for all SZA, all G-H asymmetry factors and SSA, and especially more realistic aerosol phase functions. I would recommend expanding the sensitivity studies to aerosol profiles to include 20°, 35°, 70°, 80° and 85° SZA for G-H asymmetry factors 0.64 and 0.72 and SSA 0.85 and 0.98.

2. Please discuss the effect of G-H phase function approximation on the AOD retrieval compared to a more realistic Mie phase function for different aerosol types?

3. I think that error estimation is overly optimistic especially at small SZA and small RSAA when dRSP are very small and "close" to the reference spectrum. The change in dRSP and its error do not change linearly with AMF especially for dAMF<0.5 from AMFref therefore the error in RSPref is larger then presented (0.0018). I think that more reasonable will be to either assume no RSP in the direct sun reference spectrum, or to model RSP with an RSP error equal to the RSPref itself currently derived in the paper (0.0044).

4. Method limitations need to be better stated: e.g. small AOD (how small?), clear skies (what is the tolerance to clouds), homogeneous aerosol profiles (what is the tolerance to heterogeneity), instrument FOV, instrument stray light, instrument SNR, etc.?

5. The field campaign lasted for over a month. Could the authors show all successful retrievals and show the linear correlations with other datasets based on all data not just 2 days?

Minor comments:

Line 93: described by an asymmetry factor g

Section 2.1: please describe the atmospheric conditions during TCAP in more detail (e.g cloud cover, aerosol types, vertical profiles). The authors probably have all the information to use Mie theory to calculate phase functions from other in-situ measurements.

Line 127: I suggest moving the sentences "To further... (Holben et al., 1998)" after point (3).

Line 154: What was the motivation to do almucantar scan at EA 45°. Have you analyzed these data?

Line 163: Why the authors did not use the integrating sphere to scan the sun in azimuthal and zenith direction to determine the precise position of the sun? Pointing accuracy and precise knowledge of the instrument FOV is important to characterize contribution of external stray light into the system. Please provide a figure in the supplement showing measured FOV of the instrument.

Line 180: Please clarify whether the authors use a single direct sun spectrum for the whole campaign, a single spectrum per day or for each solar almucantar scan its own DS spectrum. I believe it is crucial to have high pointing accuracy to minimize contribution of the scattered photons in the direct sunbeam measurement.

AMTD

Line 197: Why did the users use Bogumil et al., 2003 NO2 cross section compared to Vandaele et al., 1998?

Line 210: Could the authors show one figure with the dRSP error vs dSCD and one with dRSP error vs RMS, and one RMS vs RSAA for SZA 35 and 70° in the supplemental material?

Line 266: could you please specify the dates when these layers where present and the results of the AOD retrieval from the MAX-DOAS instrument? I would think that such layers indicate heterogeneity of the air masses around the observations site and potentially intervene with the retrieval.

Section 3.2: Please explore the effect of aerosol inhomogeneity on the retrieval by performing RTM simulations. Section 3.2 describes the angular asymmetry factor but does not show how it impacts the retrieval at different SZA and AOD. The authors adopt AERONET almucantar screening at 20%. But it is not clear whether this is justified for RSP measurements.

Line 532: Fig 9 shows AOD430 = 0.6 at 14:00 LST.

Line 533: I am not sure I see this. SZA at 14:00 and 11:00 LST are about the same (30°) while AOD at 14:00 is 0.6 at 11:00 is 0.3-0.4. Despite a smaller AOD (therefore larger dRSP) at 11:00 the retrieval failed. Looking at Fig S6 Asymmetry Factor Parameter is about 10% around 11:00 which might be the reason for retrieval failure.

---

## Author Comment (AC2) · 12 Apr 2016

**The CU 2D-MAX-DOAS instrument - part 2: Raman Scattering Probability Measurements and Retrieval of Aerosol Optical Properties**

Ivan Ortega[1,2], Sean Coburn[1,2], Larry K. Berg[3], Kathy Lantz[2,4], Joseph Michalsky[2,4], Richard A. Ferrare[5], Johnathan W. Hair[5], Chris A. Hostetler[5], and Rainer Volkamer[1,2]

[1]Department of Chemistry and Biochemistry, University of Colorado, Boulder, CO, USA

[2]Cooperative Institute for Research in Environmental Sciences (CIRES), Boulder, CO, USA

[3]Pacific Northwest National Laboratory, Richland, WA, USA

[4]Global Monitoring Division, Earth System Research Laboratory, NOAA, Boulder, CO, USA

[5]NASA Langley Research Center, Hampton, VA, USA

*Response to Reviewer 2; 12 April 2016*

Black: Referee's comments
Blue: Author's reply
Green: sentence added/modified in the manuscript

We thank Reviewer 2 for the helpful comments and suggestions.

Ortega et al., describe derivation of aerosol optical depth and Henyey-Greenstein asymmetry factor from solar almucantar measurements (-170_ to 170_, 5_ steps) of Rotational Roman Scattering probability. The method applies DOAS technique to hyperspectral intensity measurements to derive differential RSP in 426 – 440 nm window and therefore does not require absolute radiometric calibration. The authors present radiative transfer simulations at 430 nm using Monte Carlo model to demonstrate sensitivity of the RSP to AOD, H-G asymmetry factor, aerosol profile, relative solar azimuth angle, and solar zenith angle. They conclude that RSP is independent of the aerosol profile and has low dependence on single scattering albedo and surface reflectivity. On the other hand, RSP has high sensitivity to total AOD and H-G asymmetry factor especially at small RSAA, small AOD and large SZA. Based on these simulations they develop a method to minimize difference between retrieved (426 – 440 nm) and simulated RSP at 430 nm. Direct sun spectrum is used as a reference Fraunhofer spectrum to minimize amount of RPS in the reference spectrum. RSP in the reference spectrum is derived from Langley plot analysis of the zenith and direct sun spectra. The method is applied to 2 days, one with low and one with high AOD, during TCAP filed campaign (1 July – 13 August 2012). The retrieved AOD are compared to co-located measurements by CIMEL, MFRSR, and HSRL-2. Reasonable agreement in diurnal variability is achieved between CU 2D-MAX-DOAS, CIMEL, and MFRSR. The method is well described and the paper is well organized. I recommend publishing the paper after some modifications.

Major comments:

1. One of the main assumptions of the method is that solar almucantar measurements of RSP are independent of aerosol profiles based on the simulations at SZA 35˚ and 70˚. This might not hold for all SZA, all G-H asymmetry factors and SSA, and especially more realistic aerosol phase functions. I would recommend expanding the sensitivity studies to aerosol profiles to include 20˚, 35˚, 70˚, 80˚ and 85˚ SZA for G-H asymmetry factors 0.64 and 0.72 and SSA 0.85 and 0.98.

We have conducted simulations to show the sensitivity of AOD and aerosol extinction profile shapes at SZA of 85˚ (EA = 5˚) as suggested – Similar as Figures 4 in the manuscript and Figure S3 in the supplement. The results of this sensitivity are shown in Figure R1. As can be seen the same effects are found: high sensitivity towards AOD and low sensitivity towards aerosol extinction profile. It is quite interesting that some differences are noticeable when the aerosol extinction is aloft.

The discussion of the sensitivity towards aerosol profiles and higher SZAs (low EAs) has been expanded based on a similar comment of reviewer #1 (please see Figure R4 and the discussion in our response to reviewer 1).

[Figure]

Figure R1. Sensitivity study showing that simulated RSP (430 nm) strongly influenced by (A) AOD, and insensitive to (B) aerosol vertical distribution. (A) AOD is varied, keeping aerosols homogeneously distributed (box profile) up to 1.5 km altitude. (B) The aerosol extinction

vertical distribution is varied for a constant AOD of 0.2. The simulation is for SZA = 85˚, SSA = 0.98, g = 0.70, SA = 0.05.

Our statement were not correct for the ranges of SZA we discuss. We incorporated in the revised manuscript the next paragraph:

The sensitivity studies in Figs 3, 4, and in the supplement confirm that the RSP does not depend on the aerosol vertical distribution for SZA smaller than 80˚. Note that all of the azimuth scans here were conducted at solar EA, which for measurements at SZA < 80˚ corresponds to EAs of 10˚ or higher. For measurements at low EAs the RSP becomes slightly dependent on the aerosol vertical distribution (see Fig. S7 panel C).

We additionally incorporated Fig. R1 into Fig. S6 in the revised supplemental information. The range of 0.64 to 0.75 G-H have been shown in Figures 5 and same results are found at other SZA, which are captured by the look up table. As shown in section 2.4.2 the RSP does not show a significant variability among different SSA. Furthermore, in this study we use known SSA based on co-located observations.

2. Please discuss the effect of G-H phase function approximation on the AOD retrieval compared to a more realistic Mie phase function for different aerosol types?

We refer the Reviewer to our detailed response to Reviewer #1 (see comment A), and the new Section 3.3.3 that compares the HG phase function with Mie phase functions constrained by Aeronet observations.

3. I think that error estimation is overly optimistic especially at small SZA and small RSAA when dRSP are very small and "close" to the reference spectrum. The change in dRSP and its error do not change linearly with AMF especially for dAMF<0.5 from AMFref therefore the error in RSPref is larger then presented (0.0018). I think that more reasonable will be to either assume no RSP in the direct sun reference spectrum, or to model RSP with an RSP error equal to the RSPref itself currently derived in the paper (0.0044).

Reviewer #1 suggested a third method to quantify the RSP in the reference. We have applied this method (see revised Section 3.1), and found it supports the error bounds that we use and propagate in the paper.

We respectfully disagree that the error estimation is overly optimistic, and provide additional evidence on the robustness of the RSP fit and it's the error in RSPref below. The results support that the reported error is in fact estimated conservatively, and limited by the error in the determination of the RSP in the reference. The final error in the determination of AOD is calculated with the error propagation of the DOAS fit error and the error in the reference RSP.

Figure R2 shows RSP fit examples (top) and residuals (bottom) at SRAA of (A) 5˚ and (B) 140˚. On both examples the RSP fit error is smaller than the 0.0018 conservative error used in the manuscript. This presents additional evidence that the error in the RSP does not depend significantly in the SRAA (see also Figures R5 and R6).

Figures R3 and R4 shows the possible systematic errors quantified by changing the wavelength intervals and polynomial orders in a similar way as performed by Vogel et al. (2013) – Similar as Figure S1 and S2. From these figures it is clear that the difference with respect to the actual fitting settings are lower than 8% (RSP error of 0.0012) for SRAA of 5° and lower than 3-4% (RSP error of ~0.0015) for SRAA of 140°. Hence, the fit uncertainty for RSP is lower than the 0.0018 reported in the manuscript.

[Figure]

Figure R2. Same as Fig. 2 in the main text but for SRAA of (A) 5 and (B) 140.

[Figure]

Figure R3. Same as Figure S1 but for SRAA = 5° and SZA = 60°.

[Figure]

Figure R4. Same as Figure S1 but for SRAA = 140˚ and SZA = 60˚.

Figures R2-R4 have been added in the revised supplemental information.

4. Method limitations need to be better stated: e.g. small AOD (how small?), clear skies (what is the tolerance to clouds), homogeneous aerosol profiles (what is the tolerance to heterogeneity), instrument FOV, instrument stray light, instrument SNR, etc.?

The limitations of the method have been stated along the manuscript and in our dedicated section 3.7 "Context with literature: advantages and limitations". In the revised manuscript we estimate the response in RSP to changes in AOD based on the change of RSP with respect to AOD. In order to quantitatively show the response through different sets of AOD a linear correlation have been calculated for small subsets of AODs and the results are shown in the table R1 in our response to reviewer #1.

Regarding aerosol inhomogeneity, we have seen that the RSP is quite sensitive to such condition as discussed in the results of Fig. 9B where the retrieval method is not applied (see section 3.6). However, further investigation is needed to study in detail this conditions and the effect of broken clouds using more sophisticated simulations, if possible 3D RTM. However, this is beyond the scope of this manuscript, and mentioned in the revised manuscript.

5. The field campaign lasted for over a month. Could the authors show all successful retrievals and show the linear correlations with other datasets based on all data not just 2 days?

There are several reasons for why be believe this is neither needed, nor a good idea. We have limited this study to two days for the following reasons: (1) We are unaware of a previous attempt to retrieve AOD and g using RSP. Clear and homogeneous aerosol conditions should be the starting point to establish any new method with credibility. (2) Weather conditions during the whole deployment was characterized by overcast conditions (Berg et el., 2015), hence limiting the number of days where the evaluation of the retrieval method is straightforward; this does not

rule out that the method could not be applied to other days in the future. (3) The focus of this paper is the demonstration and evaluation of a new method, emphasize the limitations and benefits of the method including periods of large and small aerosol loading, and to validate the method with coincident independent measurements that do not operate under broken sky conditions (especially CIMEL and MFRSR – and also HSRL-2). (4) We are planning to apply the validated approach in future 2D-MAX-DOAS deployments, but there is currently a lack of suitable measurement techniques to evaluate the method under broken cloud conditions.

Minor comments:

Line 93: described by an asymmetry factor g Section 2.1: please describe the atmospheric conditions during TCAP in more detail (e.g cloud cover, aerosol types, vertical profiles). The authors probably have all the information to use Mie theory to calculate phase functions from other in-situ measurements.

In the revised manuscript we added more information about the general conditions during TCAP. We refer the Reviewer to our detailed response to Reviewer #1 regarding Mie calculations (first comment).

Line 127: I suggest moving the sentences "To further: : : (Holben et al., 1998)" after point (3).

We adopted this suggestion.

Line 154: What was the motivation to do almucantar scan at EA 45_. Have you analyzed these data?

We have not looked in detail at the almucantar scan at EA45. The advantages of evaluating azimuth scan at solar EA consist in the enhanced the sensitivity towards aerosol phase functions and minimizing the effect of aerosol inhomogeneity at small SZA. Using the EA of 45˚ would not provide fundamentally different information but could be used to check results and/or gain information of trace gases but this is beyond the scope of the manuscript. The sentence has been modified in the main text as follow:

The advantages of evaluating azimuth scan at solar EA consist in the enhanced the sensitivity towards aerosol phase functions and minimizing the effect of aerosol inhomogeneity at small SZA.

Line 163: Why the authors did not use the integrating sphere to scan the sun in azimuthal and zenith direction to determine the precise position of the sun? Pointing accuracy and precise knowledge of the instrument FOV is important to characterize contribution of external stray light into the system. Please provide a figure in the supplement showing measured FOV of the instrument.

The pointing accuracy and FOV of the 2D telescope have been characterized in detail in Ortega et al. (2015). We refer the reviewer to section 2.1.3 in that paper.

Line 180: Please clarify whether the authors use a single direct sun spectrum for the whole campaign, a single spectrum per day or for each solar almucantar scan its own DS spectrum. I believe it is crucial to have high pointing accuracy to minimize contribution of the scattered photons in the direct sunbeam measurement.

In section 2.2 we mentioned that direct sun spectra were collected for specific cloud free days. In the retrieval of the dRSP we use a single direct sun spectra under low AOD conditions (see section 2.3). The reviewer is correct, high pointing accuracy is important. As explained in section 2.3 the normalized intensities collected in every azimuth scan are used to calculate the pointing accuracy and all results shown are quality assured.

Line 197: Why did the users use Bogumil et al., 2003 NO2 cross section compared to Vandaele et al., 1998?

Thanks for catching this error. This is the wrong citation; we did use Vandaele et al. (1998).

Line 210: Could the authors show one figure with the dRSP error vs dSCD and one with dRSP error vs RMS, and one RMS vs RSAA for SZA 35˚ and 70˚ in the supplemental material?

Figure R5 shows the dRSP error vs dRSP and Figure R6 shows the dRSP error and RMS as a function of SRAA color coded by SZA. Figure 6 consolidates our conservative error reported in the manuscript and the small SRAA dependency. Both Figures have been added to the SI text.

[Figure]

Figure R5. dRSP error vs dRSP on 22 July 2012.

[Figure]

Figure R6. (top) dRSP error and (bottom) RMS vs SRAA on 22 July 2012. The gray horizontal discontinuous line represents the conservative error of 0.0018 reported in the manuscript.

Line 266: could you please specify the dates when these layers where present and the results of the AOD retrieval from the MAX-DOAS instrument? I would think that such layers indicate heterogeneity of the air masses around the observations site and potentially intervene with the retrieval.

Section 3.2 describes in more detail the aerosol inhomogeneity found in both days. The sentence has been modified slightly to point the reader out to section 3.2:

The elevated aerosol layers documented by Berg et al. (2015) during TCAP hence are captured, and do not present a limitation for this work. Section 3.2 describes in more detail the aerosol inhomogeneity on both days.

Section 3.2: Please explore the effect of aerosol inhomogeneity on the retrieval by performing RTM simulations. Section 3.2 describes the angular asymmetry factor but does not show how it impacts the retrieval at different SZA and AOD. The authors adopt AERONET almucantar screening at 20%. But it is not clear whether this is justified for RSP measurements.

The good agreement with methods that have a different field of view, and average over a different airmasses suggests that there is no limitation from aerosol inhomogeneity. We consider a systematic study that deals with aerosol inhomogeneities, their AOD and SZA dependence to be beyond the scope of this paper, which introduces a novel retrieval. We do mention about the use of 3D-RTM to assess inhomogeneous aerosol conditions and broken clouds in section 3.7.

Line 532: Fig 9 shows AOD430 = 0.6 at 14:00 LST.

Line 533: I am not sure I see this. SZA at 14:00 and 11:00 LST are about the same (30˚) while AOD at 14:00 is 0.6 at 11:00 is 0.3-0.4. Despite a smaller AOD (therefore larger dRSP) at 11:00 the retrieval failed. Looking at Fig S6 Asymmetry Factor Parameter is about 10% around 11:00 which might be the reason for retrieval failure.

We agree and meant to discuss it. The Asymmetry Factor Parameter (AFP) plays an additional important role on July 17 where values larger than 10% were identified. Section 3.2 discusses further the AFP. We believe both high AOD and high values of AFP are the reason for the retrieval to fail. The following sentence was updated in the revised manuscript:

On 17 July the AOD430 reached values of 0.6 at noon (Fig. 9B). The high AOD and the inhomogeneity identified with AFP values larger than 10% from 11:00 to 14:00 LST limited the retrieval of AOD and $g$ from the 2D-MAX-DOAS.

References

Vandaele, A. C., Hermans, C., Simon, P. C., Carleer, M., Colin, R., Fally, S., M'erienne, M. F., Jenouvrier, A., and Coquart, B.: Measurements of the $NO_2$ absorption cross section from 42 000 cm$^{-1}$ to 10 000 cm$^{-1}$ (238–1000 nm) at 220K and 294 K., J. Quant. Spectrosc. Ra., 59, 171–184, doi:10.1016/S0022-4073(97)00168-4, 1998

---

## Author Comment (AC1)

**The CU 2D-MAX-DOAS instrument - part 2: Raman Scattering Probability Measurements and Retrieval of Aerosol Optical Properties**

Ivan Ortega[1,2], Sean Coburn[1,2], Larry K. Berg[3], Kathy Lantz[2,4], Joseph Michalsky[2,4], Richard A. Ferrare[5], Johnathan W. Hair[5], Chris A. Hostetler[5], and Rainer Volkamer[1,2]

[1]Department of Chemistry and Biochemistry, University of Colorado, Boulder, CO, USA

[2]Cooperative Institute for Research in Environmental Sciences (CIRES), Boulder, CO, USA

[3]Pacific Northwest National Laboratory, Richland, WA, USA

[4]Global Monitoring Division, Earth System Research Laboratory, NOAA, Boulder, CO, USA

[5]NASA Langley Research Center, Hampton, VA, USA

*Response to Reviewer 1; 12 April 2016*

Black: Referee's comments
Blue: Author's reply
Green: sentence added/modified in the manuscript

We greatly appreciate Reviewer 1 for the review of our paper, for positive feedback, and helpful comments.

The paper by Ortega et al. describes a new method for the determination of the aerosol optical depth and information on the aerosol scattering phase function from azimuth scans of spectrally resolved observations of scattered sun light. The method has two important advantages: a) it is based on the so called Ring effect, which can be retrieved with high accuracy without the need of an absolute radiance calibration. b) the sensitivity of the method is particularly high for small AOD, for which other instruments tend to have large uncertainties.

The new method has the potential to improve AOD observations especially at low AOD. The paper is innovative, and the proposed method is well described and the first results are very promising.

I recommend publication in AMT after two major issues are addressed:

Major comments:

A) The authors describe the aerosol phase function by the Henyey-Greenstein parameterisation. This is a very simple parameterisation based on only one parameter. It is well known that true aerosol phase functions can not be well described by the HG parameterisation. Especially close to the forward direction, the deviations can become rather large. This is the range of scattering

angles, which is explored in this study. I recomment that the authors repeat their radiative transfer simulations using more realistic aerosol phase functions (e.g. using Mie phase functions based on the sun photometer measuremens as input). If this is not possible, the authors should demonstrate that the use of a HG parameterisation in their study is justified.

Thank you for this comment. We agree that the HG phase function is an approximation, and that "Especially close to the forward direction, the deviations can become rather large". However, our measurements did not probe angles smaller than 5° SRAA, where differences between HG and Mie phase functions are expected to be most visible. It is thus not correct that "This is the range of scattering angles, which is explored in this study." for reasons that we elaborate below, and in the revised manuscript.

A technical limitation exists in that our RTM only uses the HG simplification, and "realistic" phase functions are not handled. We have conducted additional sensitivity studies using RTM, in an attempt to bind the effect of Mie phase functions, and provide an explanation for the surprising fact that for the angles that were measured, the RSP observations can be reasonably well explained by the HG phase function.

The results from these calculations have been added to the revised manuscript in a new Section "3.3.3 Comparison with Mie phase function calculations". Figure R1 compares the area normalized phase functions under (A) low and (B) high AOD conditions. The red continuous lines are the retrieved $P_M(\Theta)$ reported in the AERONET web site (version 2.0) measured close in time with our RSP based retrievals. The area normalization is carried out using scattering angles of 5° and larger (i.e., 5-180°) to roughly resemble our measurements/retrieval conditions. This new Figure has been added to the revised manuscript, and results are discussed in Section 3.3.3.

[Figure]

Figure R1. Comparison of area normalized phase functions under (A) low AOD (22 July 2012 at 8:50 LST) and (B) high AOD (17 July 2012 AT 15:50 LST). The blue shaded represent a typical error in g of 10%.

The deviations between $P_{HG}(\Theta)$ and $P_M(\Theta)$ are most prominent only at small scattering angles ($\Theta < 5°$), and to a lesser extend also large scattering angles ($\Theta > 150°$), only at high AOD. For most

scattering angles, and under high and low AOD conditions, the comparison is within the 10 % error in g. We thus attribute the fact that a simplistic phase function can explain our RSP measurements reasonably well to the fact that we did not probe small scattering angles ($\Theta < 5°$).

RTM that represent Mie phase functions are desirable. However, also Mie phase functions present an approximation of the true phase function, i.e., assume particles to be spheres of a certain internal symmetry, etc. RSP measurements at scattering angles smaller 5° are potentially very interesting and hold potential to evaluate Mie theory in new ways.

B) A direct sun spectrum is used as Fraunhofer reference spectrum and the corresponding is determined by two different approaches. However, in my opinion, both approaches are based on false assumptions (see below). My suggestion would be to simply derive the RSP from radiative transfer simulations: Here I propose a simply procedure: 1) the radiance and the RSP are calculated for a scattered sun light observations in the direction of the sun. 2) the radiance of the direct sun is calculated for the same direction (the RSP for the direct light is assumed as zero). 3) The effective RSP is caluclated as the average RSP of both contributions (direct and scattered sun light) weighted by their respective radiances.

Reasons, why in my opinion both approaches in the paper are wrong:

On page 11, line 335 both methods are introduced: '(1) a Langley plot type method, where the dRSP obtained with direct sun spectra as reference spectrum is plotted as a function of the SZA, and (2) by interpolating the dRSP measured with small SRAA to the 0 degree (direct sun view).'

Method (1) is based on the assumption that the dRSP is a linear function of 1/cos(SZA). I expect that this is in general not the case.

Method (2) would only be justified if a smooth transition of the RSP between measurements of scattered and direct sun light could be expected. In my opinion this is not the case.

We have added the suggested approach as a third method to the paper. Notably, estimation of the RSP in the reference using RTM with the HG phase function ($RSP_{Ref}^{RTM}$) using steps 1 through 3 described above is not free of errors either, for reasons described under A. We have performed sensitivity studies to test the variability of $RSP_{Ref}^{RTM}$ to a range of AOD and $g$ (see Table R1).

Table R1. $RSP_{Ref}^{RTM}$ in the direct sun geometry using method 3 (SZA = 28°; SSA = 0.95).

| Run # | AOD | $g$ | $RSP_{Ref}^{RTM}$ |
|---|---|---|---|
| 1 | 0.01 | 0.68 | 0.0047 |
| 2 | 0.09 | 0.68 | 0.0040 |
| 3 (22 July conditions) | 0.1 | 0.68 | 0.0038 |
| 4 | 0.1 | 0.7 | 0.0037 |
| 5 | 0.1 | 0.85 | 0.0020 |
| 6 | 0.11 | 0.68 | 0.0037 |

| 7 | 0.2 | 0.68 | 0.0033 |
| --- | --- | --- | --- |

The weighted $RSP_{Ref}^{RTM}$ derived from this exercise for the 22 July case study is 0.0038 ± 0.0004, which compares very closely with the 0.0044 ± 0.0012 estimated from methods 1 and 2 as described in the manuscript. Interestingly, while $RSP_{Ref}^{RTM}$ is rather insensitive to AOD, a significant sensitivity exists towards a change in $g$ from 0.68 to 0.85 (roughly a factor 2). Notably, use of a Mie phase function could alter these results even more. We therefore do not believe that method 3 is better than method 1 or 2, given the current limitations of RTM.

With respect to the assumptions of method 1, we do not argue that dRSP in the reference is a linear function of the AMF. In fact, we have tested this, and it is apparent from Figure 6 that the function is nonlinear, as was described in the original manuscript. Method 1 linearizes the AMF dependence, and is fitting only over small range of AMF values.

To assess the assumptions of method 2 we have conducted further RTM simulations to test the smoothness of RSP in the transition from scattered sunlight to the direct solar beam. We have tried to circumvent the RTM limitation by approximating the Mie phase function shown in Fig. R1A with a combination of different $g$; then use these values to simulate the RSP for SRAA < 10˚ and the direct sun component. Figure R2 shows the comparison of the area normalized phase function calculated with a combination of several g ($P_{HG}^c(\Theta)$) and $P_M(\Theta)$ for scattering angles < 11˚. The $g$ values needed are also shown. Note that these $g$ values are not realistic and are used simply to approximate the results of a more realistic phase function shape over a limited range of forward scattering angles.

[Figure]

Figure R2. Comparison of area normalized phase functions calculated with a combination of g ($P_{HG}^c(\Theta)$) and $P_M(\Theta)$ by AERONET.

Figure R3 shows the RSP simulated and the corresponding effective RSP in the direct sun geometry using the $g$ values found before. Interestingly, the transition is actually quite smooth for larger $g$ and is steep only for small $g$. This can also be understood from Fig. R1.

[Figure]

Figure R3. RSP simulated for SRAA smaller than 11° and the effective RSP resulted in the direct sun geometry for several $g$ (AOD = 0.1).

We argue that method 2 is valid in the atmosphere. Whether models can be used to estimate RSP in the direct solar beam depends on the assumptions about the aerosol phase function. For HG we agree with the reviewer that a smooth transition of the RSP between measurements of scattered and direct sun light cannot be expected. However, in the atmosphere the HG is not a good approximation, and the pronounced forward scattering of a Mie phase function adds a significant weight to the RSP scattered radiance. This has the effect to smoothen the transition of the RSP between measurements of scattered and direct sun light.

Given that the results from all three methods agree within error, we have decided to describe all three methods in the revised manuscript. We have revised Section 3.1 to add the results from method 3, as well as the above rationale in support of the assumptions for methods 1 and 2. Table R1, and Figures R2 and R3 have been added to the Supplementary material. At the end of the revised section 3.5 we have added that a future deployment warrants to test the Langley plot method for more SZA to evaluate the range of SZA over which the dRSP can be linearized, and highlight the potential benefits of RSP measurements at SRAA smaller than 5° to test methods 2 and 3.

Minor comments:

Page 2, line 38: 'Active steps to minimize RSP in the reference spectrum help to reduce the uncertainty in RSP retrievals of AOD and g.'

I disgaree with this statement: Not the RSP value itself should be minimised, but rather

the uncertainty of the RSP value. In the proposed method, RSP from measurements and simulations are compared. In this comparison, the absolute, but not the relative deviation between both data sets is minimised. This means that the uncertainty of the RSP of the reference spectrum should be minimised (but not the value of the RSP itself).

We agree in that the sentence is misleading. The updated sentence reads as:

Active steps to minimize the uncertainty in the RSP help to reduce the uncertainty in retrievals of AOD and g.

Page 4, line 97: 'The quantitative analysis of RRS by DOAS was introduced by Wagner et al. (2004, 2009a) with the so-called "Raman Scattering Probability: : :'

I think additional pioneering studies should be mentoned, e.g.:

Langford, A. O., Schofield, R., Daniel, J. S., Portmann, R. W., Melamed, M. L., Miller, H. L., Dutton, E. G., and Solomon, S.: On the variability of the Ring effect in the near ultraviolet: understanding the role of aerosols and multiple scattering, Atmos. Chem. Phys., 7, 575–586, 2007.

de Beek, R., Vountas, M., Rozanov, V. V., Richter, A., and Burrows, J. P.: The Ring effect in the cloudy atmosphere, Geophys. Res. Lett., 28, 721–724, 2001.

Vountas, M., Richter, A., Wittrock, F., and Burrows, J. P.: Inelastic scattering in ocean water and its impact on trace gas retrievals from satellite data, Atmos. Chem. Phys., 3, 1365–1375, 2003.

Thanks for indicating these references. We have added them in the revised manuscript. We also added Vountas et al., 1998. The revised sentence now reads as:

Several studies have described the quantitative analysis of RRS and its effect in solar scattering UV/Vis observations (Vountas et al., 1998; de Beek et al., 2001; Vountas et al., 2003; Langford et al., 2007). The quantitative analysis of RRS by DOAS measurements was introduced by Wagner et al. (2004, 2009a) with the so-called "Raman Scattering Probability" (RSP) (the probability that a detected photon has undergone a rotational Raman scattering event). Under cloud free conditions the AOD has a strong effect on the RSP, which further exhibits a high dependency on the solar relative azimuth angle (Wagner et al., 2009b; Wagner et al., 2014). To the best of our knowledge, there has been no previous measurement of AOD and $g$ using almucantar scans of RSP by MAX-DOAS.

Page 6, line 154 : 'In this work, we focus only on the almucantar scan at solar EA.'

Why was this procedure chosen? If in addition, also measurements at other elevation angles were used (e.g. the scan at 45_ EA) additional information could be obtained.

(or the consistency of the results could be checked)

The azimuth scan at solar EA enhanced the sensitivity towards aerosol phase functions and minimizes the effect of aerosol inhomogeneity at small SZA. Using the EA of 45˚ would not provide fundamentally different information but could be used to check results and/or gain information of trace gases but this is beyond the scope of the manuscript. The sentence has been modified in the main text as follow:

The advantages of evaluating azimuth scan at solar EA consist in the enhanced the sensitivity towards aerosol phase functions and minimizing the effect of aerosol inhomogeneity at small SZA.

Page 7, line 191: RSP is not measured but retrieved

Corrected.

Page 7, line 204: 'Systematic errors in the retrieval of dRSP were quantified by means of sensitivity studies.'

Did the authors investigate the effect of instrument straylight (or its correction by the analysis software) on the RSP results?

Yes, we have investigated sensitivity towards fitting an intensity offset to correct for straylight. However, the RSP results become extremely high and nonphysical if the offset is included.

For a detailed characterization of our instruments, including the elimination of any stray light in our spectrometers, see Coburn et al. (2011).

Page 9, line 263: 'The sensitivity studies in Figs 3, 4, and in the supplement confirm that the RSP does not depend on the aerosol vertical distribution.'

This statement is based on measurements at rather high elevation angles. I expect that for measurements at low elevation angles (which are often used for MAX-DOAS observations) the RSP becomes dependent on the aerosol vertical distribution. The authors should discuss this aspect.

The method is primarily interested in the RSP at elevation angles larger than 10˚. The statement was not meant for lower EAs. Note that in the manuscript we use a fixed EA determined by the solar elevation. In order to avoid refraction a maximum recommended SZA is about 80-85˚ SZA (EA =5 -10˚). Under those conditions the RSP does not depend on the aerosol vertical distribution.

In order to assess the sensitivity of the EA scan towards aerosol extinction vertical distribution we carried out sensitivity studies. Figure R4 shows the RSP (top) and sun normalized radiance (bottom) as a function of AOD with a fixed box height of 1.5 km (Fig A and B) and fixed AOD for several aerosol vertical distributions (C and D). A greater sensitivity towards aerosol profile

shape is seen for EA smaller than 8˚. An EA split in the RSP values is only visible for EAs smaller than 5˚.

[Figure]

Figure R4. Sensitivity study showing the effect of AOD and aerosol vertical distribution on RSP and sun normalized radiance using the EA scan. The simulation is for SZA = 70˚, SSA = 0.98, g = 0.70, SA = 0.05.

The statement above has been changed to:

The sensitivity studies in Figs 3, 4, and in the supplement confirm that the RSP does not depend on the aerosol vertical distribution for SZA smaller than 80˚. Note that all of the azimuth scans here were conducted at solar EA, which for measurements at SZA < 80˚ corresponds to EAs of 10˚ or higher. For measurements at low EAs the RSP becomes slightly dependent on the aerosol vertical distribution (see Fig. S7 panel C).

Page 14, line 408: 'The highest response in RSP to changes in AOD is observed at low AOD'

I am not sure if this statement is supported by the results shown in Fig. 7. The authors should possibly use linear scales. How exactly is 'response in RSP to changes in AOD' defined

Figure S4 in the supplement shows the RSP as a function of AOD for a set of SZAs using a linear scale. In the revised manuscript we estimate the response in RSP to changes in AOD. In order to quantitatively show the response through different sets of AOD a linear correlation have been calculated for small subsets of AODs and the results are shown in the Table R1.

Table R1. Results of the linear correlation between RSP = f(AOD) for a subset of AODs using the SRAA of 5˚. The results in the table are the slope/intercept of the equation RSP = slope·AOD + intercept.

| Subset of AOD | SZA = 80° | SZA = 65° | SZA = 35° | SZA = 20° |
|---|---|---|---|---|
| 0 – 0.1 | -0.240/0.036 | -0.225/0.034 | -0.232/0.034 | -0.235/0.034 |
| 0.1 – 0.2 | -0.038/0.019 | -0.048/0.019 | -0.046/0.018 | -0.043/0.017 |
| 0.2 – 0.4 | 0.0039/0.011 | -0.013/0.012 | -0.012/0.011 | -0.013/0.011 |
| 0.4 – 1.0 | 0.018/0.004 | 0.0002/0.006 | -0.002/0.006 | -0.002/0.007 |

From this table is clear that the greatest response is at small AOD (0 – 0.1) confirming our statement in the manuscript. We believe Figure S4 shows clearly the RSP as a function of AOD with the linear scale and we decided to keep Figure 7 with the log scale and Table R1 has been added in the supplement.

Page 19, line 585: 'The error in the RSP based retrieval of AOD and g is limited by the uncertainty about RSP contained in the reference spectrum.'

I think this statement is not supported by the shown results. In particular, I think the choice of a direct sunlight Fraunhofer reference spectrum is probably not the best choice, because the instrument response (in particular instrument stray light) is probably different for scattered and direct light observations. My suggestion would be to use a Fraunhofer reference spectrum of scattered sun light from the same measurement sequence (measured in zenith direction) and to simulate instead of the RSP the dRSP.

Instrument straylight is not a limitation for our setup. In fact, adding an intensity offset during fitting give non-physical results, while the RSP fit without offset is robust. The results for AOD and g that are obtained based on RSP fits without an intensity offset further closely compare with independent measurements AOD and g by CIMEL and MFRSR. In lack of any evidence to the contrary, we respectfully disagree that the direct sunlight Fraunhofer reference spectrum is not well suited.

Using a zenith spectrum from the same measurement sequence would make the results dependent on the RSP contained in the reference spectrum. The minimization of equation 1 (see main text) would require an additional simulation of the RSP in the reference for all AODs, i.e., a separate LUT for each sequence scan due to the AOD in the reference spectrum would be unknown. We see no reason why this approach would not be equally feasible, but it is less direct than the approach chosen in this work. Use of a zenith reference may help the precision of the data, but could lead to significant offsets and limited accuracy. This text has been added to Section 3.1.1.

References

Henyey, L. G. and Greenstein, J. L.: Diffuse radiation in the galaxy, Astrophysical. Journal., 93, 70–83, doi:10.1086/144246, 1941.